



# Precision of continuous GPS velocities from statistical analysis of synthetic time series

Christine Masson[1], Stephane Mazzotti[1], Philippe Vernant[1]

Géosciences Montpellier, CNRS, University of Montpellier, Université des Antilles, Montpellier, 34000, France

Correspondence to: Christine Masson (masson@gm.univ-montp2.fr)

**Abstract.** We use statistical analyses of synthetic position time series to estimate the potential precision of GPS velocities. The synthetic series represent the standard range of noise, seasonal, and position offset characteristics, leaving aside extreme values. This analysis is combined with a new simple method for automatic offset detection that allows an automatic

treatment of the massive dataset. Colored noise and the presence of offsets are the primary contributor to velocity variability. However, regression tree analyses show that the main factors controlling the velocity precision are first the duration of the series, followed by the presence of offsets and the noise (dispersion and spectral index). Our analysis allows us to propose guidelines, which can be applied to actual GPS data, that constrain the velocity accuracies (expressed as 95% confidence limits) based on simple parameters: (1) Series durations over 8.0 years result in high velocity accuracies in the horizontal

(0.2 mm yr-1) and vertical (0.5 mm yr-1); (2) Series durations of less than 4.5 years cannot be used for high-precision studies since the horizontal accuracy is insufficient (over 1.0 mm yr-1); (3) Series of intermediate durations (4.5 – 8.0 years) are associated with an intermediate horizontal accuracy (0.6 mm yr-1) and a poor vertical one (1.3 mm yr-1), unless they comprise no offset. Our results suggest that very long series durations (over 15 – 20 years) do not ensure a better accuracy compare to series of 8 – 10 years, due to the noise amplitude following a power-law dependency on the frequency. Thus,

better characterizations of long-period GPS noise and pluri-annual environmental loads are critical to further improve GPS velocity precisions.

## 1    Introduction

GPS (Global Positioning System) and more recently GNSS (Global Navigation Satellite System) have become classical datasets to study present-day tectonics, from active plate boundary regions (e.g., Serpelloni et al., 2013; McClusky et al.,

2000; Calais et al., 2016) to intraplate domains (e.g., Frankel et al., 2011; Biessy et al., 2011; Ohzono et al., 2015). GPS data processing, and thus the associated precision of GPS velocities have significantly increased in the last 20 years owing, for example, to the contribution of studies on noise characteristics (e.g., Williams et al., 2003), ionospheric effects (Petrie et al., 2010) or multipath and geometry effects (King et al., 2010). State-of-the-art applications of GPS velocities require that the velocities be defined with higher precisions, potentially as low as 0.1 mm/yr or better. Typical examples of such

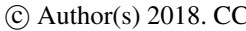



requirements are associated with debates regarding intraplate strain build-up (Calais et al., 2006; Franckel et al., 2011), regional tectonic models (Vernant et al., 2006), or fault interseismic coupling variations (Vigny et al., 2005; Métois et al., 2012).

To first order, three types of factors and processes limit the precision of GPS velocities. The first two categories are
associated with raw data processing, such as antenna phase center, satellite orbit, or atmospheric delay corrections (e.g., Tregoning and Watson 2009) and with the GPS station environment (e.g., monument stability or multipath; King and Watson, 2010). Most of these effects are difficult to assess and integrate individually in a detailed uncertainty analysis and are commonly treated as correlated noise in velocity uncertainty calculations (cf., Williams et al., 2003). The third category relates to post-processing analysis of the position time series, in particular reference frame definition (Argus et al., 1999),
periodic signals (Blewitt and Lavallée, 2002), and position offsets due to equipment modifications, earthquakes, or undefined sources (Williams, 2003a; Gazeaux et al., 2013).

The detection and correction of offsets in time series is investigated in numerous scientific domains, for example in biostatistics (Olshen et al., 2004), quantitative marketing (Fong and DeSarbo, 2007), image processing (Pham et al., 2000) or climate and meteorology (Beaulieu et al., 2008). In geodynamic GPS applications, failure to take offsets into account can
have major consequences. For example, Thomas et al. (2011) estimated velocities of about 2.1 mm yr$^{-1}$ lower than those of Argus et al., (2011), leading to very different interpretations of the data for estimating uplift rates in East Antarctica. Multiple automatic methods exist for offset detection in GPS position time series, but their reliability is limited. Gazeau et al. (2013) argue that the manual detection method is more reliable and allows the detection of smaller offsets than automatic methods, albeit with a detection rate of ca. 50%. Consequently, Gazeaux et al., (2013) consider that geophysical
interpretations of velocities smaller than 1 mm yr$^{-1}$ must be subject to particular caution, depending on the offset detection method employed.

In this study, we estimate the potential precision of GPS velocities through a statistical analysis of synthetic position time series that are representative of standard GPS data. We focus on continuous time series with a daily sampling frequency (i.e., permanent rather than campaign mode) to test the effect of colored noise, periodic signals, and position offsets (with a new
method for automatic offset detection). We illustrate our results with an application on a typical regional geodetic network in a context of low deformation (the REseau NAtional GNSS Permanent, RENAG, France, RESIF (2017)).

## 2 Synthetic time series

In order to test the factors that control the precision of velocity estimations, we simulate sets of 3600 daily position time series defined by a constant velocity, annual and semi-annual periodic motions, instantaneous offsets, and random colored
noise:

$$x(t) = vt + A_1 \sin(\omega_1 t + \phi_1) + A_2 \sin(\omega_2 t + \phi_2) + C_i.\delta(t, T_i) + D.rand(k, t) \qquad (1)$$




where the time $t$ is incremental date (with an arbitrary start at $t = 2000$); $v$ is the constant velocity throughout the whole series (set at 0 mm yr$^{-1}$); $A_{1/2}$, $\omega_{1/2}$, and $\phi_{1/2}$ are the amplitude, period, and phase of the annual and semi-annual motions; $C_i$ and $T_i$ are the amplitude and time of the i$^{th}$ offset (with $\delta$ the Kronecker function); and k is the spectral index of the colored

noise. $D$ a measure of the noise amplitude, expressed as the RMS (root-mean-square) dispersion of the position time series. Figure 1 shows an example of the decomposition of an average synthetic series.

The ranges of values of the parameters are chosen to represent the standard characteristics of horizontal and vertical components in two recent state-of-the-art GPS analyses using Precise Point Positioning and Double-Difference processing (Santamaria-Gomez et al., 2011; Nguyen et al., 2016):

- $v = 0.0 \ mm \ yr^{-1}$
- $\phi_{1,2} = 0 \ day$
- $\omega_1 = 365.25$ day
- $\omega_2 = 182.63 \ day$
- $A_1 = 1.5 \ or \ 3.0 \ mm$
- $A_2 = 0.6 \ or \ 1.2 \ mm$
- $C_i = (-6.0) - (6.0) \ mm$
- $\delta = 0 \ if \ t < T_{offi} \ or \ 1 \ if \ t > T_{offi}$
- $D = (0.6) - (4.4) \ mm$
- $k = (-0.9) - (-0.1)$
This choice of time series description and parameter values ensures a good representation of the majority of real GPS time series, but excludes both extreme parameter values (e.g., extremely noisy series) and pluri-annual or transient tectonic events such as slow slip events or post-seismic deformation (Koulali et al., 2017).

The annual and semi-annual seasonal signals have a low impact on the determination of the long-term velocity (cf., Section 3, and Blewitt and Lavallée, 2002). Because of its minor role, we only integrate the effect of seasonal signal through three

combinations of annual ($A_1$) and semi-annual ($A_2$) amplitudes (1.5 and 0.6 mm, 3 and 0.6 mm, 3 and 1.2) to illustrate first-order small, medium, and large seasonal effects on the position time series. The random noise added to the synthetic time series corresponds to the standard formula of colored noise model (Agnew, 1992):

$$P(f) = P_0 \left(\frac{f}{f_0}\right)^k \qquad (2)$$


where $f$ is the frequency, $P_0$ and $f_0$ are normalizing constants, and $k$ is the spectral index (Mandelbrot and Van Ness, 1968). We use Kasdin (1995) formulation to generate colored noise sequences characterized by their spectral indices k and the noise dispersion D of the series expressed as a Root-Mean-Square (RMS):



$$D = \sqrt{\frac{1}{N}\sum_{i=1}^{N} x_i^2} \qquad (3)$$

where N is the number of daily positions x (prior to periodic and offset integration). The chosen range of noise dispersion (0.6 – 4.4 mm) corresponds to the 90[th] percentiles of position time series in our reference studies (Santamaria-Gomez et al., 2011; Nguyen et al., 2016). Figure 2 shows the distribution of position dispersion in Nguyen et al., (2016), illustrating the

bimodal aspect of the horizontal (0.7 – 3.2 mm) and vertical (2.7 – 4.5 mm) positions.

Recent studies based on large datasets propose a range of variation of the noise spectral index $k$ between -0.8 and -0.4 (cf., Santamaria-Gomez et al., 2011; Nguyen et al., 2016). For our study, we use a slightly extended range of $k$ from -0.9 to -0.1 in order to include the effects of older noisy data (lower $k$) and hypothetical nearly-white series ($k$ close to 0). For the former, $k = -0.9$ corresponds to the average spectral index of studies on older and noisier data (Williams et al., 2004), keeping in

mind that such data can present lower $k = -1.2$ for very few series. For the latter, we consider the ongoing effort to identify, model and correct for pluri-annual climatic signal (e.g., Chanard et al., 2018), with the potential effect of "whitening" the time series by reducing the long-period amplitudes (i.e., $k = -0.1$).

Offsets in time series are defined as an instantaneous change of the position. The position and the number of offsets are chosen randomly in each series, with respectively 2, 3, 4, 5, 6, or 7 maximum offsets for time series duration 3-6, 6-9, 9-12,

12-15, 15-18, or 18-21 years, and a minimum time of 200 days between two consecutive offsets. The offset amplitude varies randomly between -6.0 and 6.0 mm with uniform distribution, excluding offsets of absolute amplitude smaller than 1.0 mm. This amplitude range corresponds to more than 80% of the values from the SOPAC archives used by Gazeau et al., 2013 and those from Nguyen et al., (2016). In the Western Europe network (Nguyen et al., 2016), the average amplitude is about 3.0 mm with a standard deviation of 3.0 mm. Although extreme values can reach ca. 10 and 25 mm for the horizontal vertical

components, we limit our synthetic range to ± 6.0 mm in order to stay within the time series dispersion (i.e., extremely large offsets are as easily detected and corrected as large ones).

Figure 3 shows a variety of synthetic position time series illustrating the quality of the data used in our study. In these different examples we can already identify for which parameters or combinations of parameters it will be most difficult to determine the long-term velocity (fixed at 0 mm yr$^{-1}$). For series with the same duration a high noise (k, D) and the presence

of offsets seem to hinder the determination of the long-term velocity. In the rest of the study, we will quantify these different effects.

## 3    Effect of parameters on the velocity accuracy

In this section, we analyze the effect of each model parameter (independently and combined) on the velocity estimation. For each time series, all the parameters are jointly estimated using a linear least-square inversion of the model (Eq. 1), except for





the noise parameters that are estimated independently using a spectral analysis of the residual time series (Eq. 2). The results are analyzed using several statistical indicators of the absolute values of the estimated velocities. Because the true velocities are set to 0.0 mm yr$^{-1}$, these statistics represent the deviations of the velocity estimations from the expected values, and are thus a measure of the accuracy of the estimated velocities. The various analyses are presented in whisker plots, from which

we derive two main indicators: $v_{95}$, the velocity that corresponds to the 95 percentile (i.e., the 95% confidence limit in the velocity accuracy), and $p_{01}$, the percentile associate with a velocity accuracy of 0.1 mm yr$^{-1}$. We use regression tree analyses (Breiman et al., 1984) to hierarchize the role, defined as the importance (Ishwaran et al., 2007), of the parameters on the velocity accuracy.

The impact on velocity estimations of seasonal signals and offsets alone (without added noise) is extremely limited. A

simple linear model including only a long-term velocity and either annual / semi-annual sinusoid or Kronecker delta functions can be inverted to retrieve the exact parameter values, provided that the time series is long enough (at least ca. 3 years) and that it is not affected by several offsets at very near positions (few days apart). Simple tests performed by inverting such series confirm this hypothesis by yielding velocity accuracies of 0.01 mm yr$^{-1}$ for the shortest series (< 4 yr) and better than 0.01 mm yr$^{-1}$ in all other cases, including any of the three combinations of annual and semi-annual seasonal

terms. Thus, in the following we focus on the effect of colored noise alone and colored noise with offsets, which are the main contributors to the velocity uncertainties.

### 3.1    Effect of colored noise

In order to estimate the impact of colored noise alone, we construct synthetic series using a subset of Eq.1:

$$x(t) = vt + D.rand(k,t) \qquad (4)$$

We first analyze the effect of the three parameters - the duration of the series ($T$), the spectral index ($k$) and the noise dispersion ($D$) - independently of the others. Figure 4 shows the velocity accuracies as a function of these three parameters. The worst values of velocity accuracy due to noise alone can reach $v_{95}$ = 0.7 mm yr$^{-1}$ for the shortest series (T < 5 yr). For

series longer than 15 years all $v_{95}$ are smaller than 0.1 mm yr$^{-1}$, with a possible asymptotic value ca. $v_{95}$ = 0.05 mm yr$^{-1}$. A near-exponential decrease of the $v_{95}$ velocity accuracy is observed as a function of the duration of the series. The dependence of the velocity accuracies on noise parameters ($k$ and $D$) shows an expected accuracy decrease with smaller spectral indices (closer to -1) and higher noise amplitudes, with a near-exponential increase with $D$. Overall, the percentile of velocity accuracies smaller than 0.1 mm yr$^{-1}$ is $p_{01}$ = 86%. The 14% of series with accuracies larger than 0.1 mm yr$^{-1}$ are associated

with the shortest and noisiest series.

A joint analysis of the parameters using a regression tree confirms their relative importance, with the most important being the series duration $T$ (56%) followed by the spectral index $k$ (35%) and the noise dispersion $D$ (9%). Figure 5 shows the tree classification (Fig. 5a) and the whisker plots of the associated leaves (Fig. 5b). The branches and the associated leaves are




ordered in order of importance and leaf size from left to right. Hereafter, we limit the tree classification to three node levels in order to only highlight the primary controlling elements.

The tree classification shows that a velocity accuracy of $v_{95}$ = 0.1 mm yr$^{-1}$ is achieved for over 2/3 of the series (leaves 1 and 2) corresponding to all the long series (T > 11.0 yr, leaves 1 and 2) and those with average durations and large spectral

indices (6.1 < T < 11.0 yr, k > -0.6, leaf 1). The overall velocity accuracies degrade for the other leaves. A velocity accuracy $v_{95}$ = 0.2 mm yr$^{-1}$ is still reached for combinations of average durations and small spectral indices (6.1 < T < 11.0 yr, k < -0.6, leaf 3) or short durations, large spectral indices and low noise amplitude (T < 6.1 yr, k > -0.7, D < 2.6 mm, leaf 4). The remaining cases (short duration, small spectral index, high noise) represent less than 10% of the samples and result in poor accuracies of $v_{95}$ = 0.4 mm yr$^{-1}$ (leaf 5) and $v_{95}$ = 0.7 mm yr$^{-1}$ (leaf 6).

Additionally, a significant piece of information emerging from the regression tree analysis is the relatively low coefficient of determination $R^2$ ~ 0.5, which indicates that the combinations of the three model parameters (T, k, D) only explain about 50% of the observed velocity dispersion. This points out the strong effect of the stochastic noise generation, which alone accounts for about half of the velocity variability. In other words, for a given set of parameters, the generated time series will show variable characteristics (noise structures) that randomly impact the velocity estimations. We illustrate this point by

estimating the velocity dispersion for a sample of 300 series with constant parameters T = 10 yr, k = -0.7, D = 3.0 mm (belonging to leaf 3 of the tree). The estimated velocities show a RMS dispersion of 0.2 mm yr$^{-1}$, of the same order as the velocity dispersion observed in leaf 3 (Fig. 5b). This effect is all the more important, as the series is short.

As noted in the introduction to Section 3, seasonal signals have very little effect on the velocity estimations. This is also true for seasonal signals added to series with random noise, which yield similar results to those presented above for noise alone

(e.g., $p_{01}$ = 86%) with the seasonal parameters ($A_{1/2}$ combinations) ranking with negligible importance in the tree classification (less than 1%).

### 3.2  Effect of offsets

In order to test and estimate the effect of position offsets on velocity estimations, we analyze synthetic time series that include offsets added to seasonal and random noise (Eq. 1). This choice is justified by the very low effect of offsets alone

(cf. introduction of Section 3) and the fact that this combination is representative of real data, thus providing useful estimations of the expected accuracy of actual velocities. In the case of real data, dealing with offsets required either fixing their dates (from equipment logs or earthquake catalogs) or detecting their potential occurrences. In section 4, we will come back to how to consider the latter. For this section, we quantify the two end-member cases in which we either do not know and therefore do not solve any offset, or we know and solve all of them.

#### 3.2.1  Effect of unresolved offsets

In this first simple case, we test time series with a single offset that is not solved, and quantify the importance of the offset parameters (amplitude $C_1$ and position in the series $T_1$) in addition to the parameters T, k, and D considered previously. A




regression tree analysis indicates that the velocity variability is primarily controlled by the time series duration $T$ (importance 49%), as in the case of noise alone, followed closely by the amplitude of the offset (40%). The position of the offset (5%), the noise amplitude (3%) and spectral index (3%) rank in 3[rd], 4[th], and 5[th] positions far beyond the two main parameters. The coefficient of determination is larger than for the noise alone ($R^2 = 0.8$), indicating that the inclusion of a single offset

contributes significantly to the overall velocity variability.

This is illustrated in Figure 6, which shows a distribution of velocity accuracies much larger than for the noise alone (cf. Fig. 4), with $v_{95}$ systematically above ca. 0.3 mm yr$^{-1}$. The presence of a single unresolved offset increases the $v_{95}$ to 0.5 mm yr$^{-1}$ for long series ($T > 13$ yr) and up to 2.5 mm yr$^{-1}$ for short series. Only about 1/5 of the series are associated with velocity accuracies below 0.1 mm yr$^{-1}$ ($p_{01} = 18\%$, compare to $p_{01} = 84\%$ for noise-alone series). As expected, the position of the

offset in the series has a significant impact, with an offset placed at one end of the series causing a velocity deviation much lower than an offset placed in the central part.

In a second series of tests, we include, but do not solve, several offsets (between 0 and 7 offsets depending on the series length, cf. Section 2). In this case, we cannot quantify the impact of the amplitudes and positions of the offsets as single parameters; instead we use the ratio number of offsets to the series duration T, which illustrates the proportion of offsets in

the series. A regression tree analysis indicates the following parameter importances: T (53%), ratio of number of offsets to T (44%), D (2%) and k (1%), similar to the case of a single shown above. About 2/3 of the series are associated with velocity accuracies below 0.1 mm yr$^{-1}$ ($p_{01} = 67\%$). The largest velocity deviations occur on the shortest series.

Uncorrected offsets are therefore a dominant element in the quality of the long-term velocity. These conclusions on the role of the position and magnitude of the offsets in the time series are consistent with the analytical analysis in Williams (2003b).

### 3.2.2   Effect of resolved offsets

As in the previous section, we first analyze the simple case of a series with one offset, but for which we fix the date and solve for the amplitude during the inversion. Thus, the velocity accuracies are affected by the possible imperfection of the estimated amplitude of the offsets, primarily due to the series colored noise. The regression tree analysis indicates that, when the offset amplitude is solved, the offset parameters become of very low importance (amplitude and position at 2% each),

while the series duration and noise parameters recover the same importance and order as in the case of noise alone: $T$ 52%, $k$ 31% and $D$ 13% (cf. section 3.1). The regression tree and associated velocity accuracy statistics are similar to that of the noise alone analysis (cf. Fig. A in appendix). The $v_{95}$ accuracies are approximately 3 times lower than those in the case of an unresolved offset but slightly larger than those in the case of noise alone, in particular in the case of short series.

Considering series with a variable number of offsets, for which we fix the date and solve for the amplitude, the importance

of the parameters becomes intermediate between the noise-alone and single-offset cases: T 42%, ratio of number of offsets to T 21%, k 20% and D 17%. Resolving the offset amplitudes reduces their importance (21 % vs. 44%) but their presence remains a significant source of velocity variability, contrary to the case of a single solved offset by series. This is readily explained by the fact that the offset amplitudes are not perfectly resolved due to complex interaction between the offset





positions, their amplitudes, and the noise structure that result in potentially very short linear segments in the series. This is illustrated by the percentile of series with accuracies lower than 0.1 mm yr$^{-1}$ ($p_{01}$ = 71%), slightly lower than in the case of noise only series ($p_{01}$ = 86%).

This latest result represents the maximum possible accuracy of series with several offsets, assuming that all offset dates are

know. In reality, we do not know the exact nature and dates of all potential offsets (e.g., Gazeau et al., 2013), so it is necessary to detect them before solving for their amplitude. In the next section, we propose a new detection method and test its impact on velocity accuracies.

## 4    A new approach for offset detection and impact on velocity accuracy

### 4.1    Methodology of offset detection

Real GPS time series are associated with an indeterminate number of offsets, which are classically included as instantaneous changes of position in the series inversion; cf. Eq. 1 where offsets are modeled as $C_i \cdot \delta(t, T_i)$ where $C_i$ and $T_i$ are the amplitude and time of the i$^{th}$ offset (with $\delta$ the Kronecker function). The dates $T_i$ can be based on equipment logs, catalogs of earthquakes, or routines that detects offsets in the position series (Gazeau et al., 2013). Here we propose a slightly different approach that does not consists in seeking where there are offsets, but rather in seeking where there are none.

This simple principle is implemented by defining artificial offset dates that are regularly spaced in the series every $\Delta d$ days. The series is then inverted to estimate all offset amplitudes ($amp_{off}$) and their associated standard errors ($\sigma_{off}$) jointly with the other model parameters (velocity, seasonal signal, etc.). A simple significance test is then performed on the offset of smallest amplitude:

$$|amp_{off}| \geq b * \sigma_{off} \quad (5)$$

If the amplitude is larger than its scaled standard error, the offset is considered significant. Because the test is performed on the smallest offset and the offset standard errors are similar in the majority of cases, we then consider that all offsets are significant and keep them in the model. In the opposite case (amplitude smaller than the scaled standard error), the offset is

rejected and the model is rerun to test the new smallest offset, until a significant offset is found.

This very simple method can be simply implemented in most time series analysis and only requires and empirical calibration of the two parameters $\Delta d$ and $b$. After several tests, we set the former to $\Delta d = 20\ days$, which corresponds to the lower limit before the method breaks down (i.e., too many undifferentiated offsets). The latter is set to $b = 20$, which allows a good compromise between the detection of real offsets defined in the synthetic series and the detection of false positives (cf.

Section 4.2). Details on the parameter calibration and the detection levels are available in appendix B.



### 4.2 Detection ability

By applying our method to series with only one offset, it is possible to determine the conditions of offset detections. Overall, 67% of the offsets are detected. The detection capacity depends primarily on the duration of the time series $T$, combined with the series noise amplitude $D$ and the offset amplitude $C$. For the shortest time series ($T < 6$ yr), we detect 21% of offsets.

They correspond to the series with the largest offsets ($C > 3.0$ mm) and the smallest noise amplitudes ($D < 2.1$ mm). There is no offset detection in the series with large noise amplitude ($D > 2.1$ mm).

For the time series of 6 to 18 years, we can detect offsets of small amplitudes ($C = 1 – 3$ mm) in series with low noise levels ($D < 2.1$ mm) and large offsets ($C > 4$mm) in all series. For the longest time series of more than 18 years, one widens still the range of detection. Offsets larger than 3 mm are all detected and those of between 2 and 3 mm are detected at 49%. The very

small offsets ($C < 2$mm) are detected only in the low noise series ($D < 2.1$ mm).

By applying our method on the sample with several offsets, the detection ability is decreased due to offset and noise interactions. Overall, the performance level is characterized by ca. 52% of true detections (and so 48% of missed detections) of the theoretical total number of offsets and about 20% of false positives (cf. appendix B for detection calibrations). These statistics are similar or slightly better than those of the most efficient automatic and manual detection methods analyzed in

Gazeaux et al., (2013). Although not perfect, our method allows us to obtain robust and quantitative results, and is suitable for processing of very large datasets such as our synthetic series or regional and global massive processing efforts that become increasingly common (e.g., Kreemer et al., 2014) and that could not be analyzed "by hand".

### 4.3 Impact on the determination of the velocities

The application of the offset detection method on a full dataset with multiple offsets, variable noise and seasonal signals

provides a sample that can be considered as close as possible to actual GPS data. We use this analysis to provide constraints on potential velocity accuracies in "real" data. Overall, nearly 2/3 of series are associated with velocity accuracy smaller than 0.1 mm yr$^{-1}$ ($p_{01} = 61\%$). This is lower than in the cases of noise alone ($p_{01} = 86\%$) or of fully resolved offsets ($p_{01} = 71\%$), but significantly better than in the case of unresolved offsets ($p_{01} = 33\%$). The difference between the detection method and the fully resolved results (ca. 10%) is mainly associated with undetected offsets.

For the regression tree analysis, the integration of a parameter associated with offsets is complex. Although these parameters (numbers total of offsets, of true and false detections, positions in the series, amplitudes) are known in our synthetic data, this is no the case in actual datasets. Tests on several offset parameters indicate the total number of offsets in the series ($N_{off}$) is both the simplest and the one with the highest prediction capacity. This new regression tree (Fig. 7) confirms the major role of the series duration ($T$ 55%) and noise dispersion ($D$ 16%) in explaining the variability of the velocities, but the total

number of offsets now take the second position ($N_{off}$ 25%), above the noise dispersion. It is particularly worth noting that number of offset is in fact a binary predicator (splitting value $N_{off} = 0.5$) corresponding to either the absence ($N_{off} = 0$) or the presence ($N_{off} \geq 1$) of offsets in the series. To first order, the regression tree results can be divided in three categories:

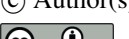


The best velocity accuracies ($v_{95} \sim 0.2 - 0.3$ mm yr$^{-1}$) are associated with either long (T > 8.0 yr) and low noise dispersion (D < 2.3 mm) series, or with series of intermediate duration (4.5 < T < 8.0 yr) with no offset (leaves 1 and 3). These represent over 42% of the dataset.

Intermediate accuracies ($v_{95} \sim 0.5 - 0.6$ mm yr$^{-1}$) are associated with series characterized by long duration and high

dispersion series (leaf 2), intermediate duration and low dispersion (leaf 4), or short duration (T < 4.5 yr) but no offset (leaf 6). Altogether, these represent another 43% of the dataset.

The remainder ca. 15% correspond to poor accuracies ($v_{95} > 1.0$ mm yr$^{-1}$) and is mostly associated with short durations (leaves 7, 8, 9), or intermediate duration and high dispersion (leaf 5).

Tree nodes associated with the series dispersion $D$ indicate that a systematic separation can be made at D = 2.2 – 2.3 mm

(Fig. 7a). As shown in Figure 2, the separation between horizontal and vertical dispersion occurs ca. D = 2.5 mm, close to the node splitting value. Thus, we can consider that the node split based in the series dispersion represent a first order distinction between (mostly) horizontal and vertical GPS components, although noisy horizontal and very clean vertical data can obviously be positioned in different categories.

On these bases, a fairly simple set of rules can be derived from the regression tree analysis that may be applicable to actual

GPS data, considering the fact that series duration is the key parameter:

A duration of 8.0 years or more ensures a high accuracy in both horizontal ($v_{95} = 0.2$ mm yr$^{-1}$) and vertical ($v_{95} = 0.5$ mm yr$^{-1}$) components.

Short series with less than 4.5-year duration cannot be used for high-precision studies ($v_{95} > 1.0$ mm yr$^{-1}$), except in the rare cases when one can be certain that they contain no significant offset.

For intermediate durations (4.5 < T < 8.0 yr), only series with no offset can provide a high accuracy ($v_{95} = 0.3$ mm yr$^{-1}$). All others are associated with an intermediate horizontal accuracies ($v_{95} = 0.6$ mm yr$^{-1}$) and a poor vertical one ($v_{95} = 1.3$ mm yr$^{-1}$).

The strong dependency on the absence or presence of one or more offsets in intermediate and short series corresponds to the effect described in Section 3.2 and confirms that the resolution of the offset amplitude if limited by the complex interactions

between offsets and noise structures. This effect is very strongly reduced (or possibly suppressed) when offsets affect long (T > 8.0 yr) series. For those, the velocity variability is independent of offset presence (Fig. 7a) because such series will maintain relatively long "offset free" segments that ensure a good resolution of the velocity parameter.

Finally, it is significant that no tree node exists that distinguishes very long series. In other words, the effect of the series duration is limited to ca. 4.5 yr and 8.0 yr. This is consistent with the observation made in the noise-alone analysis that the

decay of the noise effect as a function of time stagnates ca. 15 to 21 years at $v_{95} \sim 0.05$ mm yr$^{-1}$ (cf. Fig. 4 and section 3.1). These results may indicate a lower limit in velocity accuracy ca. 0.1 mm yr$^{-1}$ due to the colored nature of the time series noise. In other words, longer series may not ne able to significantly improve the velocity accuracy without additional efforts to whiten the noise through better data processing (cf. section 1) or taking into account pluri-annual signals. However, this hypothesis is only valid under the simple noise model (linear spectra, Eq. 2) used in our synthetic data. Alternative noise





models exist that suggest a flattening of the spectra at long periods (e.g., Gauss-Markov model, Langbein et al., 2004), which would strongly limit the pluri-annual effect and allow a much stronger impact of long series duration. The actual nature of GPS noise at periods longer than 5 – 10 years is poorly defined (Santamaria-Gomez et al., 2011; Hackl et al., 2011) and is thus a major unknown in analyses of velocity accuracies.

### 4.4 Validation of velocity standard errors

For each series, the velocity standard error is calculated using Williams (2003) generic expression for colored noise with non-integer spectral index. In order to estimate the spectral index and amplitude of the colored noise, we use a simplified least-square inversion in which we fit a linear model to the series power spectrum limited to periods between 1/12 and T/2 years (with T the length of the time series). In contrast with a more complex non-linear method, such as maximum

likelihood, this simple approach does not solve for the noise crossover frequency and thus only provides a first-order estimate of the noise parameters and velocity standard errors. We can test these standard errors against the expected result ($v$ = 0.0 mm yr$^{-1}$) in our synthetic time series to estimate their robustness.

Figure 8 shows the distributions of the ratio of the velocity accuracy to its standard error ($v / \sigma_v$). A ratio of 1 corresponds to a standard error equal to its velocity. A ratio smaller (greater) than 1 corresponds to a standard error greater (smaller) than its

velocity. Owing to our stochastic approach, and assuming a Gaussian distribution of the velocities and standard errors, we can consider that, if the standard error calculations where correct, ca. 68% of the ratios should be smaller than 1 (i.e., 68% of the velocities are included in their standard errors) and ca. 95% of the ratios should be smaller than 2 (i.e., 95% of the velocities are included in twice their standard errors).

In our dataset, only 54% are smaller than 1 and 75% are smaller than 2 (Fig. 8). These percentages are low and suggest that,

on average, our velocity standard errors are too small by a factor of about 1.6. This result is primarily controlled by the series spectral index, while the series duration and dispersion have little effect (Fig. 8). Series with indices ca. (-0.6 > k > -0.9) are associated with ratio percentages close to the 68 and 95% marks. In contrast, series with high indices (k > -0.6) present ratios that are too low especially for very high indices (k > -0.4). These results suggest that the simplified (linear spectra) approach yields reasonable results for series with near-flicker (k < -0.6) noise characteristics, but significantly underestimates the

standard errors for series with near-white (k > -0.4) noise.

### 5 Application to the RENAG data

The statistical analyses of synthetic data presented in the previous section provide guidelines to estimate the precision of velocities from actual GPS time series. In particular, the regression tree classification based on the full synthetic dataset and automatic offset detections (section 4.3) can be used to associate any actual time series with a specific velocity precision

(e.g., $v_{95}$) simply using the series parameters (duration, noise amplitude and spectral index). In this section, we present an example of such an application on the French RENAG network.





### 5.1 Offsets due to equipement changes

The RENAG network comprises 74 stations whose equipment modifications are fully documented (cf., http://webrenag.unice.fr), thus providing a good test case for our offset detection method. On the 222 time series (i.e. 3 components times 74 stations) with durations between 2.0 and 18.4 years, the comparison of detected offsets with the station

logs show that a change of receiver is very rarely associated with an offset (only 6% of the 137 cases), whereas a change of antenna causes an offset almost systematically (75% of the 8 cases) with average amplitudes of 2 – 3mm in the horizontal and 13 mm in the vertical components, respectively. However, these percentages are not robust due to the small sample sizes (especially the antenna changes). A more robust analysis would require a larger dataset, as well as the distinction between equipment changes within large data gaps or near the ends of the time series. Additionally, the offset detection method could

be improved to integrate the probability that an offsets occurs on all three component of the same station, rather than individually as it is currently done.

### 5.2 Potential velocity precision of the RENAG stations

The data from the 74 RENAG stations come from a Precise Point Positioning solution combined with noise reduction using a regional stack techniques (Masson et al., 2018; Nguyen et al., 2016). The time series of each station position component

(north, east, up) are treated independently. Based on their durations ($T$), numbers of offsets ($N_{off}$), dispersions ($D$) and spectral indices ($k$), the series are associated with one of the leaf of the regression tree (Fig. 7a) and with the corresponding $v_{95}$ (Fig. 7b). This provides an estimation of the velocity precision similar to a 95% confidence interval. Note that in this case, we use the term "precision" as it applies to actual GPS velocities for which the "true" value is not known. We also consider that the number of detected offset is similar to the total number of offsets ($N_{off}$), assuming that undetected offsets

have small amplitudes and a small impact on the velocity estimations. This hypothesis is problematic for short series where the detection capacity is low (cf. Section 4.2) and it is likely that offsets were not detected and that it leads to the misclassification of series in the leaf 6.

Figure 9 shows a map of the RENAG stations with the $v_{95}$ precision of each component according to the tree leaves. Roughly half (53%) of the 74 stations are associated with the highest precisions in the horizontal (north and east, $v_{95}$ = 0.2 mm yr$^{-1}$)

and vertical ($v_{95}$ = 0.5 mm yr$^{-1}$) components. In a few cases (12%), the east component is degraded to a slightly larger precision $v_{95}$ = 0.5 mm yr$^{-1}$. About 1/3 (30%) of the stations correspond to cases with no detected offsets and identical precision in all three components, either $v_{95}$ = 0.3 mm yr$^{-1}$ or $v_{95}$ = 0.6 mm yr$^{-1}$ depending on the duration of the time series.

Recent studies of GPS data in Western Europe have shown tectonic signals at the limit of GPS resolution. The most significant signal corresponds to a systematic uplift of 1 – 2 mm yr$^{-1}$ in the central and northern regions of the Western Alps

(Brockman et al., 2012; Nguyen et al., 2016; Nocquet et al., 2016). The pattern of uplift and its lateral variations can provide important information on the associated dynamic (e.g., postglacial rebound versus subduction tear, Chéry et al., 2016; Nocquet et al., 2016). Our analysis suggests that the 95% confidence level of the RENAG velocities in the Alps is ca. 0.5



mm yr$^{-1}$, which may still be too large to provide strong constraints on the dynamic processes. In parallel with the vertical signal, horizontal deformation is starting to emerge in the GPS data analysis that show radial extension rates ca. 0.2 – 0.5 mm yr$^{-1}$ in the Western Alps and Pyrenees (Nguyen et al., 2016; Rigo et al., 2015; Walpersdorf et al., 2018). Such rates are at the limit of the 95% precision estimated for individual RENAG stations (Fig. 9). This is especially true of stations in the

French Jura, which shows a relatively low precision $v_{95}$ ~ 0.6 mm yr$^{-1}$ due to their recent installation and short time series (T < 3.5 yr). These examples highlight the importance of network redundancy and high station density in order to strengthen the deformation analysis by relying on several nearby stations to reduce aleatory noise in individual GPS time series.

## 6    Conclusions

We used statistical analyses of synthetic position time series to determine the potential accuracy of continuous GPS
velocities. Our results are representative of standard GPS time series, leaving aside cases with extreme noise levels (e.g., random-walk) or transient tectonic signals (e.g., slow slip events).

In the synthetic datasets, time series random noise combined with the presence of position offsets are the primary contributors to the variability of the estimated velocities, whereas seasonal signals have a negligible effect. Using regression tree analyses, we show that the duration of the time series is the main parameter controlling the data classification and the
velocity precision. It is followed by the absence / presence of at least one offset and by the series dispersion due to random noise. Within the range of tested values, the nature of the random noise (near-white to near-flicker) does not contribute to the velocity variability at significant level.

We derive a set of guidelines, which can be applied to actual GPS data, that provide constraints on the velocity accuracy due to simple time series parameters (i.e., duration, presence of at least one offset, and noise dispersion; cf. Fig. 7). The velocity
accuracies are given as the 95 percentile in each class ($v_{95}$), akin to a 95% confidence limit:

Series with a duration of 8.0 years or more are associate with a high velocity accuracy in the horizontal ($v_{95}$ = 0.2 mm yr$^{-1}$) and vertical ($v_{95}$ = 0.5 mm yr$^{-1}$) components, regardless of their other characteristics (offset presence, nature the noise).

Series with a duration of less than 4.5 years cannot be used for high-precision studies ($v_{95}$ > 1.0 mm yr$^{-1}$), except when they are not affected by any offset ($v_{95}$ = 1.0 mm yr$^{-1}$ horizontal and vertical).

Series of intermediate duration (4.5 – 8.0 years) and no offset can provide a high accuracy ($v_{95}$ = 0.3 mm yr$^{-1}$). Those, more common, with at least one offset are associated with an intermediate horizontal accuracy ($v_{95}$ = 0.6 mm yr$^{-1}$) and a poor vertical one ($v_{95}$ = 1.3 mm yr$^{-1}$).

A significant outcome of our analysis is the fact that very long series durations (over 15 – 20 years) do not ensure a better accuracy compare to series with 8 – 10 years of measurements. This effect derives directly from our noise model definition,
in which the noise amplitude follows a classical power-law dependency on the frequency (Eq. 2; Agnew, 1992). As a result, the noise amplitude constantly increases with long periods, explaining the very small effect of the time series duration past ca. 10 years (cf. Fig. 4). Alternative noise models, such as Gauss-Markov, that predicts a flattening of the power spectrum at





long periods would likely change our results and reinstate a significant duration dependency for very long series. This shows the importance of better characterization of the GPS noise nature at very long periods and of current efforts to model and correct for long-period signals such as pluri-annual environmental loads.

## 5 Acknowledgments

We are grateful to Pr. Gilles Ducharme (IMAG, U. Montpellier) for his critical help with the regression tree analysis. The synthetic datasets and statistical analyses were performed using R (R Core Team, 2016).

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



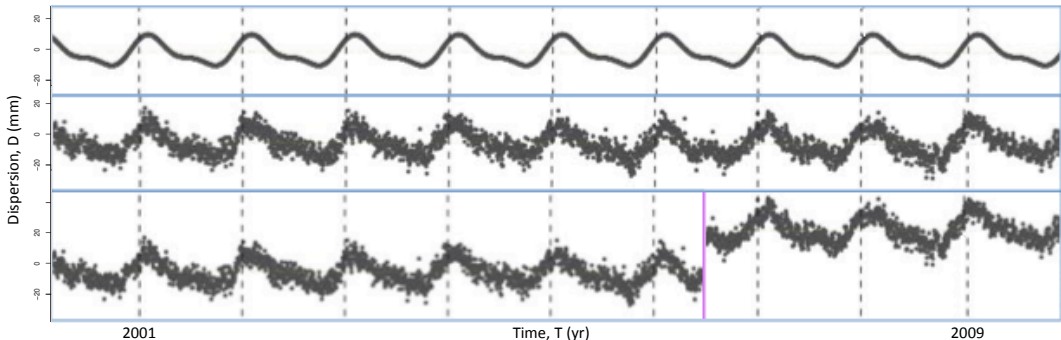

**Figure 1: Decomposition of an example synthetic time series. Top: seasonal (annual and semi-annual) signals. Middle: seasonal signals combined with random colored noise. Bottom: seasonal signal, colored noise, and a simulated offset (vertical pink line).**

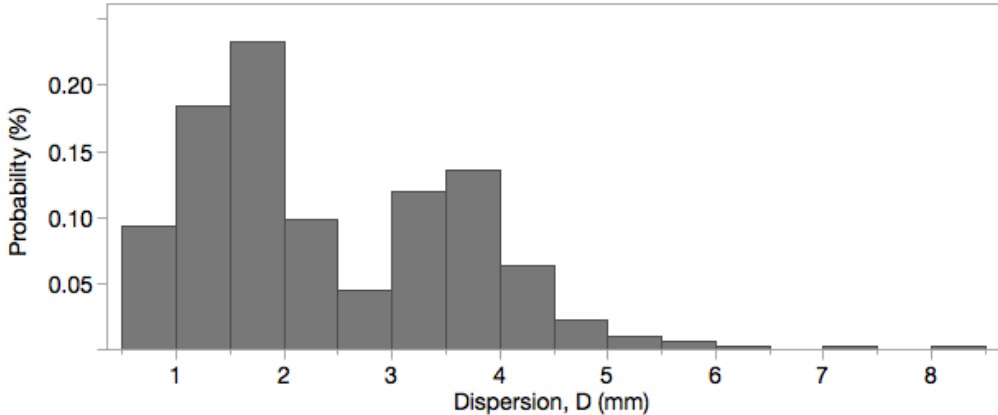

5   **Figure 2: Distribution of position dispersion (measured as RMS) in Nguyen et al. (2016) solution for Western Europe with bimodal aspect of the horizontal (0.7 – 3.2 mm) and vertical (2.7 – 4.5 mm) positions.**





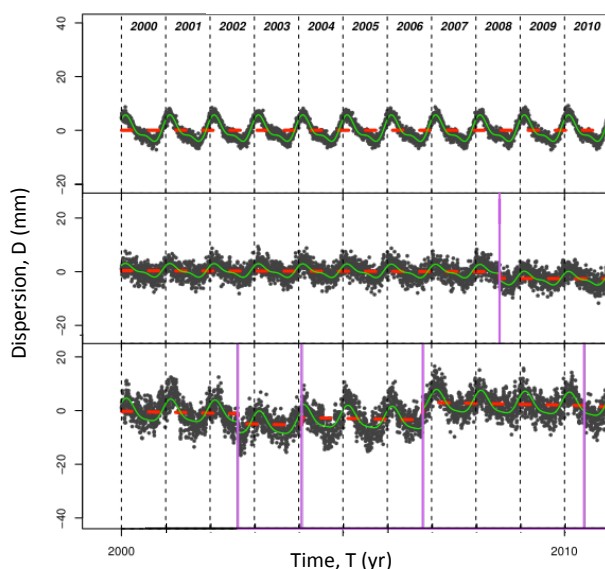

**Figure 3: Examples of synthetic time series. Each black dot represents a daily position. In green, the modeled seasonal signal, in red the linear velocity trend and the pink vertical lines represent the offsets. The 3 examples illustrate the quality of the data used in our study. Top: a slightly noisy serie (k = 0.3, D = 1.2 mm) without offset. Middle: a moderately noisy serie (k = 0.4, D = 2.3 mm) with 1 offset. Bottom: a noisy serie (k = 0.7, D = 3.5 mm) with multiple offsets.**

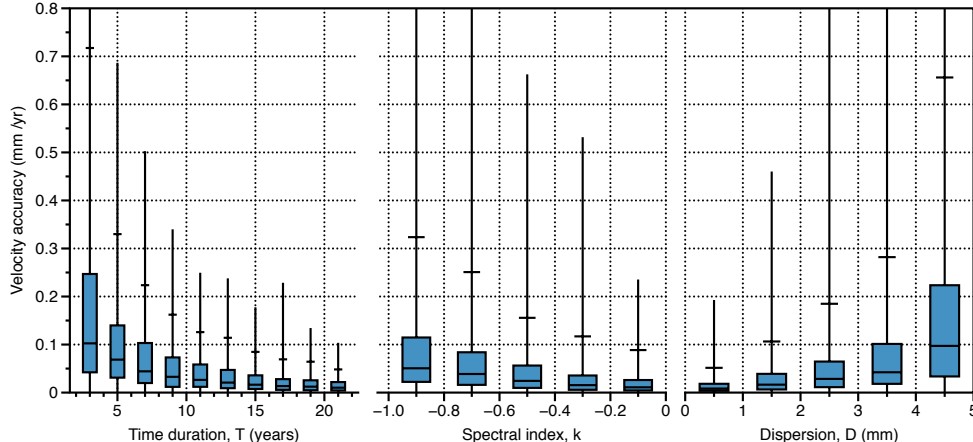

**Figure 4: Whisker plots of velocity accuracy as a function of the duration (T), spectral index (k) and dispersion (D) for series including only colored noise. The horizontal top bar represents $v_{95}$.**





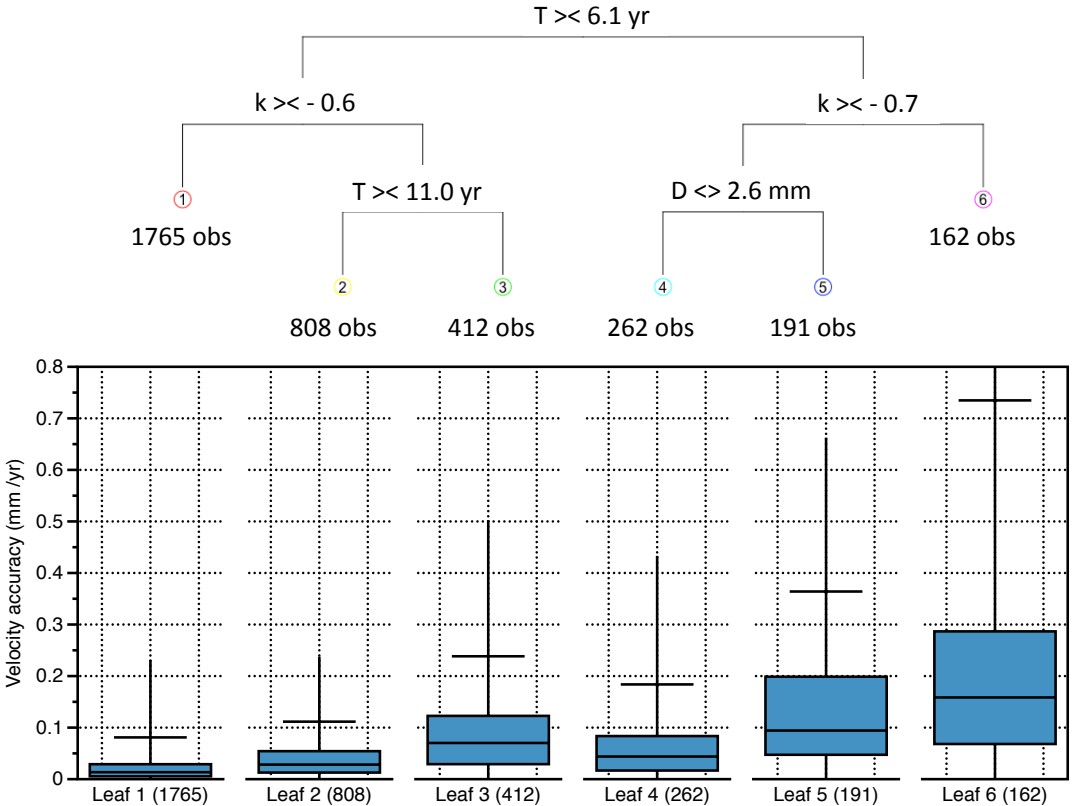

**Figure 5: The regression tree classification (Fig. 5a) and the whisker plots of the associated leaves (Fig. 5b) for series including only colored noise. The horizontal top bar represents $v_{95}$.**

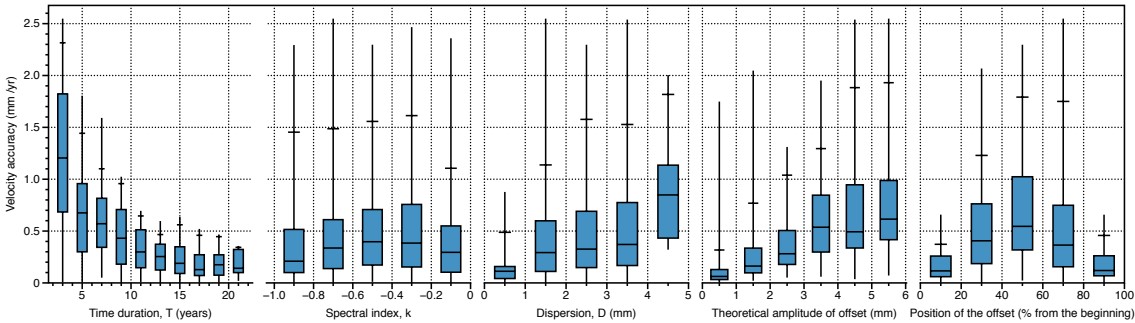

5  **Figure 6: Whisker plots of velocity accuracy as a function of the duration (T), spectral index (k), dispersion (D), offset amplitude ($amp_{off}$) and offset position in percentage for series with only 1 offset unresolved. The horizontal top bar represents $v_{95}$.**





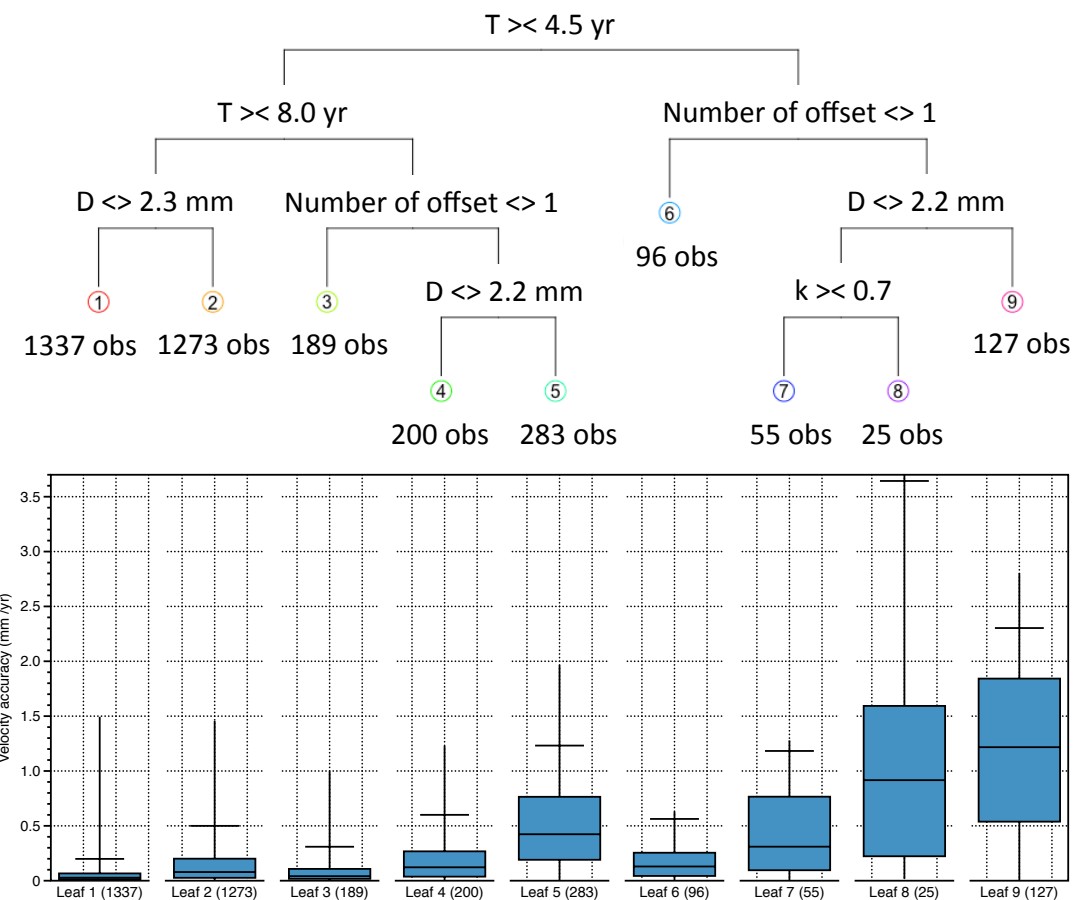

**Figure 7: The regression tree classification (Fig. 7a) and the whisker plots of the associated leaves (Fig. 7b) for the full dataset with multiple offsets, variable noise and seasonal signals with application of the offset detection method. The horizontal top bar represents $v_{95}$.**

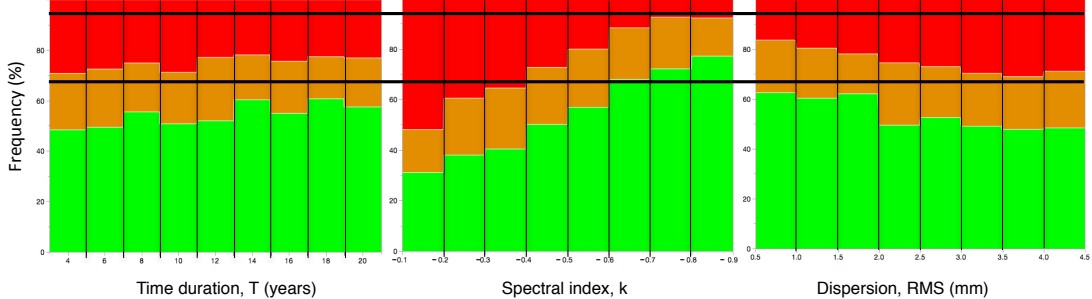

**Figure 8: Distributions of the ratio of the velocity accuracy to its standard error ($v / \sigma_v$). A ratio of 1 corresponds to a standard error equal to its velocity. A ratio smaller (greater) than 1 corresponds to a standard error greater (smaller) than its velocity. Ratio: less than 1 in green, less than 2 in orange and greater than 2 in red. The black lines correspond to the 68 and 95% confidence interval of the standard errors.**





**Figure 9: Map of the RENAG stations with the $v_{95}$ precision of each component according to the tree leaves.**



**A Effect of resolved offsets**

This figure shows the regression tree for Section 3.2.2 about the effect of several resolved offset.

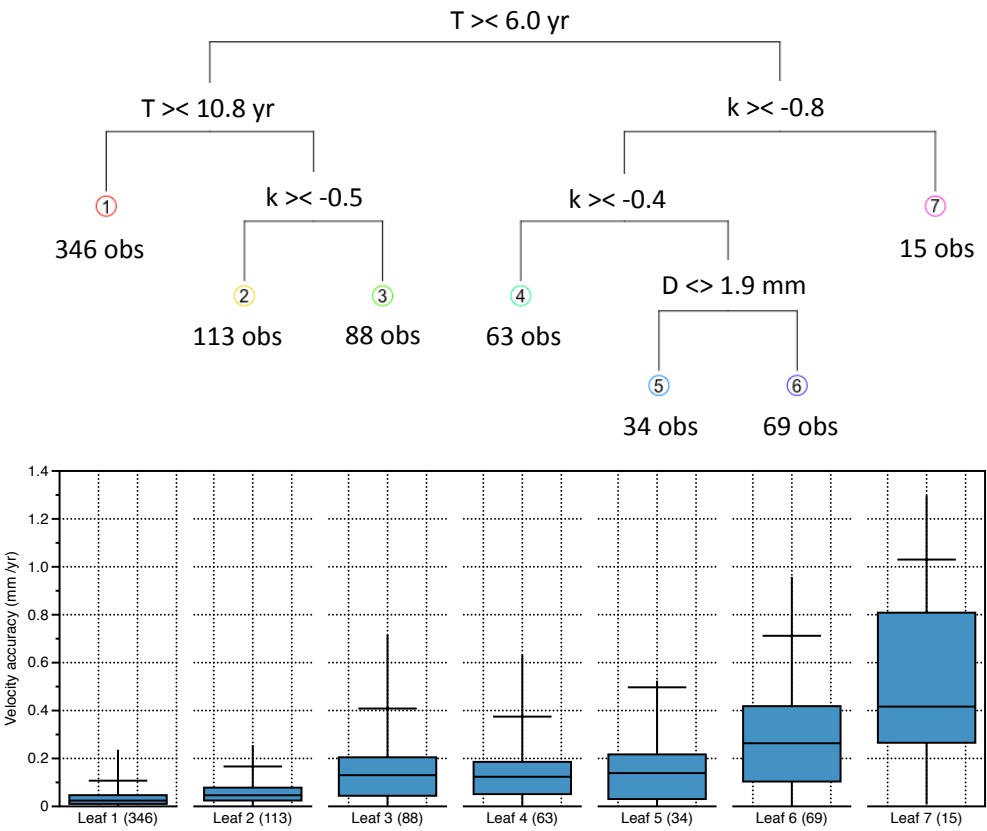

5    **Figure A:  The tree classification (Fig. Aa) and the whisker plots of the associated leaves (Fig. Ab) for the full dataset with multiple resolved offsets, variable noise and seasonal signals.**

**B Methodology of offset detection**

In order to choose the optimum values of the parameters $b$ (significance threshold) and $\Delta d$ (offset separation date), we used a subset of data (450 series) representative of the characteristics of section 2. We tested 9 combinations of parameters with $b =$

10    15, 20, 25 and $\Delta d = 10, 20, 35$. Then we compared the percentage of offsets detected, relative to the total number of offsets,





as well as the number of false detections (i.e., no actual offset), relative to the total number of detections (Fig B). A first remark is related to the calculation time required for the $\Delta d = 10$ analyzes, which is 10 times higher than for $\Delta d = 20$. In addition, the results obtained are only slightly better for false detections but lower for the percentage of true detections. Analyzes with $b = 25$ or $\Delta d = 35$ clearly give poorer results. The results of the combinations $b = 15 - \Delta d = 20$ and $b = 20 - \Delta d$

5   $= 20$ show very similar results (difference $< 1\%$), we finally chose the combination $b = 20 - \Delta d = 20$. A test at $b = 10$ showed a strong increase of the false detections, so we decided not to apply it to the entire dataset.

Regarding the false detections, we have several indicators allowing us to discern part of them, considering that they are commonly associated with a very large uncertainty and are easily identifiable, but this is not the case for all. They are mainly related to the noise structure (small k), which can create several false detections very close together. There remains a part of

10   false detections not discernible.

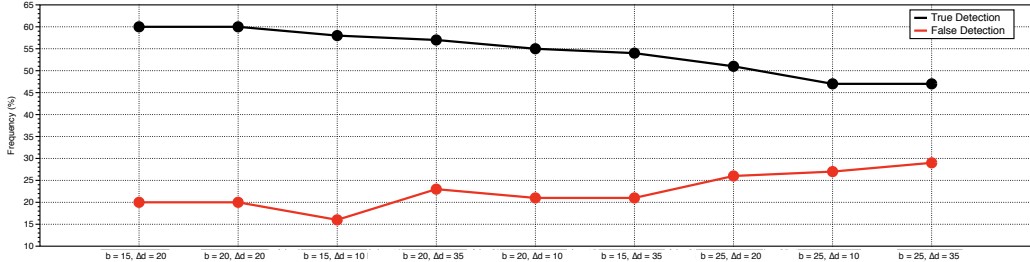

**Figure B: Percentages of true and false detections based on combinations of *b* and Δ*d*.**