# Peer review of "Precision of continuous GPS velocities from statistical analysis of synthetic time series"

_Solid Earth, 2018_

## Referee Comment (RC1) · S. Williams (Referee) · 16 Oct 2018

When assessing trends from GPS (or indeed other) time series it is very hard to understand what competing factors have the most influence of the trend and especially its uncertainty. These factors can be such things as the amplitude and severity of the time-correlated noise in the series, the presence of periodic signals, the influence of offsets, whether detected or not, or simply the length of the time series. Many of these influences have been dealt with separately but very few, if any, have attempted to capture the combined effects from all the factors and derive a metric/methodology for categorizing the severity of each effect. This paper attempts to do this using synthetic series, which have the same characteristics as real GPS time series derived from previous papers that have looked at the various effects individually. The authors have come up

with quite a reasonable and simple set of metrics to categorize a time series. Overall I think this paper is a worthwhile addition to the "error analysis" body of evidence in GPS time series estimation and will help steer other groups to understanding the limits of their GPS time series in order to be neither over optimistic or pessimistic in their assessment of the uncertainties of their results.

The only real question I have is in simulating the offsets did the authors choose a minimum time span of 200 days? They could have followed the same methodology as in Gazeaux et al [2013] and chosen a binomial distribution with a probability of 1 in 950 which will give about the same number of offsets per number of years but will not restrict the offsets to occur more than 200 days apart. In addition the DOGEX dataset is a great dataset that has been used by many authors to check their offset estimation algorithm against other solutions. Since this dataset also has very similar properties to the those created in this paper it would have been good for the authors to have tried their method out on the DOGEX dataset just as a standard against which to compare.

Technical issues.

There are a few places in the paper where the authors say serie instead of series. Also I am not familiar with the notation used in the regression tree plots but I guess I understand what <> and >< mean but it would probably be good to mention somewhere what they mean.

---

## Referee Comment (RC2) · Hammond (Referee) · 31 Dec 2018

General Comments:

Masson et al. present an analysis of constructed position time series that were made using analytical forms of signal and noise that are typically observed in geodetic GPS data. They then use these synthetic time series to identify the factors that contribute the most to uncertainty in the estimated trend. Once the most important factors are identified (time series length, spectral content and amplitude of colored noise, etc.), they offer a few rules of thumb that can be applied to categorize time series according to how precise they are expected to be. One of their conclusions is that the time series duration is invariably the most influential factor in maintaining low uncertainty in velocity

estimation, which is very important when considering how GPS networks are funded and maintained.

Aside from the focus on synthetic time series, two elements in the paper stand out as looking new to me. First, they employ a regression-tree approach that rank orders parameters in terms of their overall impact on the velocity uncertainty. This is useful since these factors are sometimes known for real data in advance and can be used to generate expectations for which time series provide the lowest uncertainties, before any more detailed analysis is undertaken. Second they introduce a new method for scanning the time series for discontinuities that are undocumented, i.e., their existence and time of occurrence are unknown beforehand. Their method is interesting because they flip the problem by determining which epochs contain *no* step to within data uncertainty, thereby narrowing the set of epochs that could have steps.

I had a number of suggestions, mostly minor which I placed in the technical comments below. The introduction could benefit from some short additional text, possibly in the last paragraph, on the general value of looking at synthetic time series as opposed to 1) real ones when so many are available, or 2) simple formulas that mathematically represent the content of signals+noise in them (e.g., Williams, 2003). The answer might be e.g., the ability to know the true answer in order to evaluate the validity of false and true detections, which might be obvious at the onset to expert readers but not everyone. That paragraph would be a good place to also mention the limitations of an analysis like this, since many real GPS time series contain signals of types not included in their synthetic tests. They mention a few examples in the paper but do not discuss the impact of the potential presence of these signals in detail.

Some detailed/technical comments:

Page 2 line 26. Could replace "low deformation" with "low rate of deformation"

Page 2 line 31. In equation 1 they may have meant to use H(t) rather than the Kronecker delta function to indicate the occurrence of a step in the time series. H(t), the Heaviside

function, is zero before t and one after t, and is also the time integral of the Dirac delta function. The Kronecker delta function is a discrete version of the Dirac delta function, in physics literature. Here Masson et al., define delta(t) in a way that works for their paper so it is probably all OK and self-consistent here, but might cause some minor confusion to call it the "Kronecker delta function".

Figure 2. It seems odd to me at first to lump all the horizontal and vertical data together in the analysis, and in this one plot. I guess it all works out in the end. But I wondered if including a new binary parameter in the regression tree, horizontal vs. vertical time series, would have a strong predictive ability in the tree.

Figure 3. The caption lists values of k as positive, but on page 3 they are said to be always negative.

Page 5 line 26. It is a little confusing sometimes that they interchange the terms "accuracy" and "uncertainty". For example, Figure 4 is a nice plot, but "accuracy" should be changed to "uncertainty" since accuracy should improve (increase) with time series duration, but the quantity shown decreases with time series duration. Also in Figure 4, they should state in the caption what is the meaning of the vertical extent of the vertical black bar, and also what is indicated by the extent of the blue box. Then in 5, 6, 7 it can be said they are as in Figure 4. These plots may be standard in some literature but probably not everyone will already know the details of construction.

Page 5 line 25. "possible asymptotic value ca. v95 = 0.05 mm yr-1". Possibly? In Figure 4 it looks like the uncertainty is still decreasing, though more slowly, at duration 20 years. I would have thought that theoretically the asymptote would be v95=0 and maybe we will do at least a little better if we run a GPS station for 100 or 1000 years. I don't see evidence of an asymptote at 0.05 mm yr-1.

Figure 8. I think in the caption v should be v_95? The "95" is dropped in several places when it should be included.

Page 6 line 18. Probably meant "if the series is short."?

Page 8 line 11. it would be better to use a lower case t, rather than T for the step time so as not to confuse with time series duration.

page 8 line 14. "consists in" should be "consist of"

Page 8 line 16. Instead of "amp_off", notation for non-offsets might be better stated in same class as true offsets, e.g., C_notanoffset or something shorter.

Page 8 line 25. I have a few questions about their interesting new offset detection method. First, it seemed that a part of the explanation may be missing. It is stated that it is repeated "until a significant offset is found". But once an offset is found there could be others and, since only evenly spaced arrays of offsets are tested in each iteration, there may be epochs that have not yet been tested. So how does the algorithm guarantee the completeness of the scan for a step at every epoch? Secondly, if a large true step exists and the adjacent epoch is tested, it will likely be evaluated as a significant step. Is there a mechanism to replace the adjacent epoch with the correct one once it has been tested? When the process is repeated are steps and non-steps identified in previous iterations excluded from being considered as steps? If so I expect that would improve efficiency and reduce ambiguity in the algorithm. Finally, could this method be applied to real data? It seems that the calibration method determining b and delta_t in Appendix B relies on the quantity of false and true identifications, so might not be available for real data(?).

Page 9 line 21. "real" need not be in quotes.

Page 9 line 27. "is no the case" should be "is not the case"

Page 10 line 14-22. These classifications may be useful but possibly a bit dismissive of the utility of some of the categories when signals are large enough to stand out from the noise. For example the Oregon coast rises >4 mm/yr owing to elastic strain accumulation on the subduction zone. Even short time series may be useful there.

Page 10 line 31. "These results may indicate a lower limit in velocity accuracy ca. 0.1 mm yr-1". But it said in the previous sentence that some were 0.05 mm yr-1...

Page 11 line 13. "v" should be "v_95"? Also "A ratio of 1 corresponds to a standard error equal to its velocity". In a Gaussian distribution +/- one standard deviation contains 68% of the samples, whereas the definition of v_95 in this paper is the limit that contains 95%. So would not a ratio of 2 indicate that the standard error and velocity accuracy are similar?

Page 12 line 14. I did not see Masson et al., 2018 in the reference list.

Page 13 line 28. "A significant outcome of our analysis is the fact that very long series durations (over 15 – 20 years) do not ensure a better accuracy compare to series with 8 – 10 years of measurements". However, Figure 4 says they are still getting more precise even at 20 years (though apparently at a decreasing rate of improvement) so I'm not sure if this statement is strictly true. It may be true that if a specific requirement for uncertainty is 0.1 mm/yr then there is no need to collect longer time series, but that requirement standard depends on the application and we may not yet know all future standards that are needed from the data.

Page 14 line 5. Acknowledgements sections often now contain proper attribution to those who collected (in this case the RENAG network), archived, processed the data, and from where the processed time series were downloaded, i.e ftp server, web site, etc., and on what date. In this case the authors may have had prior access to the data (?), i.e. processed it themselves, but it would improve repeatability of this work if others could be guided to where they could access the data. Separate questions: Are the synthetic time series developed here openly available?

Page 25, line 5. Why not show b=10, discussed in the text, on the plot?

---

## Author Response (AR1)

**RC1. Simon Williams**

When assessing trends from GPS (or indeed other) time series it is very hard to understand what competing factors have the most influence of the trend and especially its uncertainty. These factors can be such things as the amplitude and severity of the time- correlated noise in the series, the presence of periodic signals, the influence of offsets, whether detected or not, or simply the length of the time series. Many of these influences have been dealt with separately but very few, if any, have attempted to capture the combined effects from all the factors and derive a metric/methodology for categorizing the severity of each effect. This paper attempts to do this using synthetic series, which have the same characteristics as real GPS time series derived from previous papers that have looked at the various effects individually. The authors have come up with quite a reasonable and simple set of metrics to categorize a time series. Overall I think this paper is a worthwhile addition to the "error analysis" body of evidence in GPS time series estimation and will help steer other groups to understanding the limits of their GPS time series in order to be neither over optimistic or pessimistic in their assessment of the uncertainties of their results.

The only real question I have is in simulating the offsets did the authors choose a minimum time span of 200 days? They could have followed the same methodology as in Gazeaux et al [2013] and chosen a binomial distribution with a probability of 1 in 950 which will give about the same number of offsets per number of years but will not restrict the offsets to occur more than 200 days apart.

Effectively, Gazeaux's approach is more detailed than ours and allows for a more complete coverage of possible offset positions. However, in contrast with the study of Gazeaux et al (2013), our primary objective is not to study offset detection and characterization, but their impact on the velocity estimation. A random generation of offset dates can result in series with several offsets separated by only a few days or weeks. In our analysis, tests showed that such series are treated as an equivalent series with a single offset that corresponds to the combination of the 2 or 3 nearby offsets. As a result, such series will bias the general statistics by assigning single-offset statistics to multiple-offset datasets. Thus, we chose a separation of 200-days or more as a simple approach to both see the effect of offsets over short periods (a few months) and without disrupting the statistics with consecutive offsets (a few days). More detailed analyses with more random offset position distributions could be performed, but we are quite confident that the overall patterns and statistics would be similar. This is illustrated indirectly by our offset detection method, which shows that the exact offset date (within ± 10 days) does not significantly impact the results.

Page 5 Line 6: years. We use a minimum time lapse of 200 days between two consecutive offsets. Although not realistic, this lapse of time avoids distorting the overall statistics with consecutive offsets that are treated to a single offset in our detection method (cf. section 4).

In addition the DOGEX dataset is a great dataset that has been used by many authors to check their offset estimation algorithm against other solutions. Since this dataset also has very similar properties to the those created in this paper it would have been good for the authors to have tried their method out on the DOGEX dataset just as a standard against which to compare.

Indeed it would be a very good idea to apply our methods to the DOGEX dataset. We did not do it because the offset detection method is not the core of our study and article. Even though the taking into account of offsets is a major aspect in the determination of the long-term velocity, our main objective is to quantify the importance of each factor and to obtain an estimate of the possible bias according to the characteristics of the series. The offset detection method that we developed here has many possible improvements as well as developments that are currently under consideration for future work (at which time we would like to have the chance to use the DOGEX dataset to compare and understand the results). In parallel, the French RENAG working group is working on a comparison of different time series analysis methods (including the one developed here). The preliminary results show that our offset detection method performs as well as others. More detailed discussion could be considered with the group from Gazeaux et al. (2013).

**Technical issues.**

There are a few places in the paper where the authors say serie instead of series.

**We made the necessary changes.**

**Figure 3:** Examples of synthetic time series. Black dots represent daily positions. Green, red, and pink links show modeled seasonal signal, velocity and offsets. The 3 examples illustrate the quality of the data used in our study. Top: a slightly noisy series (k = -0.3, D = 1.2 mm) without offset. Middle: a moderately noisy series (k = -0.4, D = 2.3 mm) with 1 offset. Bottom: a noisy series (k = -0.7, D = 3.5 mm) with multiple offsets.

Also I am not familiar with the notation used in the regression tree plots but I guess I understand what <> and >< mean but it would probably be good to mention somewhere what they mean.

**We added an explanation of this notation:**

Page 6 Line 22: ... from left to right. The comparison signs (> <) or (<>) are relative to each tree separation, with the sign on the left corresponding to the left branch and the sign on the right corresponding to the right branch. Hereafter,...

**RC2. William Hammond**

**General Comments:**

Masson et al. present an analysis of constructed position time series that were made using analytical forms of signal and noise that are typically observed in geodetic GPS data. They then use these synthetic time series to identify the factors that contribute the most to uncertainty in the estimated trend. Once the most important factors are identified (time series length, spectral content and amplitude of colored noise, etc.), they offer a few rules of thumb that can be applied to categorize time series according to how precise they are expected to be. One of their conclusions is that the time series duration is invariably the most influential factor in maintaining low uncertainty in velocity estimation, which is very important when considering how GPS networks are funded and maintained.

Aside from the focus on synthetic time series, two elements in the paper stand out as looking new to me. First, they employ a regression-tree approach that rank orders parameters in terms of their overall impact on the velocity uncertainty. This is useful since these factors are sometimes known for real data in advance and can be used to generate expectations for which time series provide the lowest uncertainties, before any more detailed analysis is undertaken. Second they introduce a new method for scanning the time series for discontinuities that are undocumented, i.e., their existence and time of occurrence are unknown beforehand. Their method is interesting because they flip the problem by determining which epochs contain \*no\* step to within data uncertainty, thereby narrowing the set of epochs that could have steps.

I had a number of suggestions, mostly minor which I placed in the technical comments below.

The introduction could benefit from some short additional text, possibly in the last paragraph, on the general value of looking at synthetic time series as opposed to 1) real ones when so many are available, or 2) simple formulas that mathematically represent the content of signals+noise in them (e.g., Williams, 2003). The answer might be e.g., the ability to know the true answer in order to evaluate the validity of false and true detections, which might be obvious at the onset to expert readers but not everyone. That paragraph would be a good place to also mention the limitations of an analysis like this, since many real GPS time series contain signals of types not included in their synthetic tests. They mention a few examples in the paper but do not discuss the impact of the potential presence of these signals in detail.

**We added the following elements to address this point:**

Page 1 Line 27: ... However, several state-of-the-art applications of ... be defined with increasingly better precisions, ...

Page 2 Line 23: In this study, we estimate the potential precision of GPS velocities through a statistical analysis of synthetic position time series that are representative of standard GPS data. We focus on continuous time series with a daily sampling frequency (i.e., permanent rather than campaign mode) to test the effect of colored noise, periodic signals, and position offsets (with a new method for automatic offset detection). The use of synthetic data allows a detailed analysis of the velocity estimations compared to the target ("true") velocities and of the specific contribution of each parameter that can be treated independently. On the contrary, such an analysis would not be possible with real GPS data in which the true value and role of each parameter cannot be fully deconvolved. The parameter range used in the synthetic data is representative of typical average data and excludes the potential effect of transient phenomenon, such as slow slip or postseismic events, or that of pluri-annual hydrological processes. The impact of such phenomenon is addressed in several recent studies (e.g. Altamimi et al., 2016; Chanard et al., 2018) and could be included in more detailed synthetic analyses beyond our present study. We illustrate ...

**Some detailed/technical comments:**

Page 2 line 26. Could replace "low deformation" with "low rate of deformation"

**We made the necessary changes. Thank you. (Page 2 Line 32)**

Page 2 line 31. In equation 1 they may have meant to use H(t) rather than the Kronecker delta function to indicate the occurrence of a step in the time series. H(t), the Heaviside function, is zero before t and one after t, and is also the time integral of the Dirac delta function. The Kronecker delta function is a discrete version of the Dirac delta function, in physics literature. Here Masson et al., define delta(t) in a way that works for their paper so it is probably all OK and self-consistent here, but might cause some minor confusion to call it the "Kronecker delta function".

**Yes indeed, changed that. Thank you for pointing this mistake.**

Figure 2. It seems odd to me at first to lump all the horizontal and vertical data together in the analysis, and in this one plot. I guess it all works out in the end. But I wondered if including a new binary parameter in the regression tree, horizontal vs. vertical time series, would have a strong predictive ability in the tree.

The main difference between horizontal and vertical time series is the noise level. The ranges of values overlap between horizontal and vertical noise levels, so we prefer not to make a distinction and to carry out the analysis without testing for "horizontal series vs. vertical series". As the reviewer indicates, this "works out in the end" since the noise levels acts as an indicator that separates the first-order horizontal and vertical data, without forcing an a priori distinction, which is quite remarkable.

Figure 3. The caption lists values of k as positive, but on page 3 they are said to be always negative.

**Indeed, we made the necessary changes. Thank you.**

Page 5 line 26. It is a little confusing sometimes that they interchange the terms "accuracy" and "uncertainty". For example, Figure 4 is a nice plot, but "accuracy" should be changed to "uncertainty" since accuracy should improve (increase) with time series duration, but the quantity shown decreases with time series duration.

We agree that this is confusing... We used the term "accuracy" because it corresponds to what we measure, i.e. the deviation of the estimated velocity from the true velocity. Hence a "high accuracy" in common phrasing actually corresponds to a small number (small deviation from the true value). Using "uncertainty" would help (i.e., "high uncertainty" = large number), but it leads to the confusing situation pointed out by the reviewer in which both terms are used but are not interchangeable. In the original manuscript, we also used the term "precision" as a generic word to be more in line with classical studies of GPS velocity uncertainties, which discuss "precision" (dispersion around a central value) and not "accuracy" (which cannot be known for actual GPS data).

In order to clarify this, we propose to replace the term "accuracy" by "bias", which corresponds to the same concept (deviation from true value) but has an inverse amplitude, i.e. "high bias" = large number. Thus in Figure 4 (and others), the increase in the time series duration leads to a decrease in the bias associated with a smaller number. We also added a couple of sentences to explain this (cf. P2L23, comment above).

**We added the following elements to address this point:**

**Page 3 Line 1:**

*Velocity bias* – For each time series, the calculated velocity is compared with the true (imposed) velocity. The absolute value of the difference between the two is termed "velocity bias" and represents the deviation of the calculated velocity compared to the truth. We choose the term "bias" rather than "accuracy" in order to avoid confusion (e.g., a high accuracy associated with a small number) and different definitions of "accuracy". For each analysis, the velocity bias distribution is characterized by two statistical estimators:

95% confidence limit (noted  $v_{95}$ ) – This estimator is the 95% quantile of the bias distribution and represents a 95% confidence in the estimated velocities.

*Probability of 0.1 mm yr*-1 (noted  $p_{01}$ ) – This estimator is the percentile associate with a velocity bias of 0.1 mm yr-1. E.g.,  $p_{01} = 75\%$  indicates that a 75% probability that the velocity bias be smaller than or equal to 0.1 mm yr-1.

*Precision* – We limit the usage of the term "precision" to the general concept of "quality" of a velocity estimation, regardless of its origin and whether it corresponds to a systematic error (bias) or a measurement repeatability (dispersion).

*Standard error* and *Uncertainty* – For each time series, the calculated velocity and other parameters are associated with standard errors estimated as part of the linear inversion (cf. Section 3). These standard errors are used as estimators of the uncertainty on each calculated velocity.

Also in Figure 4, they should state in the caption what is the meaning of the vertical extent of the vertical black bar, and also what is indicated by the extent of the blue box. Then in 5, 6, 7 it can be said they are as in Figure 4. These plots may be standard in some literature but probably not everyone will already know the details of construction.

Caption of Figure 4: ... noise. Whiskers diagrams show the data quartiles (25, 50, 75%) in blue, the extremes (0%, 100%) with the vertical black line, and the 95 percentile (v95) with the horizontal black line.

Page 5 line 25. "possible asymptotic value ca. v95 = 0.05 mm yr-1". Possibly? In Figure 4 it looks like the uncertainty is still decreasing, though more slowly, at duration 20 years. I would have thought that theoretically the asymptote would be v95=0 and maybe we will do at least a little better if we run a GPS station for 100 or 1000 years. I don't see evidence of an asymptote at 0.05 mm yr-1.

Yes indeed, the use of the word asymptote is unjustified. You are absolutely right.

Page 6 Line 12: For series longer than 15 years, all v95 are smaller than 0.1 mm yr-1. A nearexponential decrease of v95 is observed as a function of the duration of the series with a sharp slowdown from 15 years of data.

Figure 8. I think in the caption v should be v\_95? The "95" is dropped in several places when it should be included.

**No, it is actually the velocity bias of each individual time series (and not the 95 percentile of the full dataset). This has been clarified in the caption:**

... Distribution of the ratio of the velocity bias to its standard error for each individual time series.

Page 6 line 18. Probably meant "if the series is short."?

**Yes, exactly. Thank you.**

Page 7 Line 8: This effect is more important if the series is short.

Page 8 line 11. it would be better to use a lower case t, rather than T for the step time so as not to confuse with time series duration.

**Yes, thank you for this idea. In the text we have replaced $T_i$ with $t_i$ .**

Page 8 line 14. "consists in" should be "consist of"

**Yes, exactly. Thank you. (Page 9 Line 6)**

Page 8 line 16. Instead of "amp\_off", notation for non-offsets might be better stated in same class as true offsets, e.g., C\_notanoffset or something shorter.

**The notation was confusing. We clarify it to use the same notation as the time series equation (eq. 1) to show that we are testing the artificial offsets added:**

Page 9 Line 8: ... The series is then inverted to estimate all offset amplitudes ( $C_i$ ) and their associated standard errors ( $\sigma_{ci}$ ) jointly with the other model parameters (velocity, seasonal signal, etc.). The offset with the smallest amplitude ( $C_s$ ) is then identified and a simple significance test is performed:

**$|C_S| \geq b \cdot \sigma_{cs}(5)$**

If the amplitude ( $C_S$ ) is larger than its scaled standard error ( $b.\sigma_{cs}$ ), the offset is considered significant. Because the test is performed on the smallest offset and the offset standard errors are similar in the majority of cases, we then consider that all offsets are significant and we keep them in the model. In the opposite case, the smallest offset is rejected and the inversion is redone with the remaining offsets in order to test the new smallest offset, until a significant offset is found or none remains. Page 8 line 25. I have a few questions about their interesting new offset detection method. First, it seemed that a part of the explanation may be missing. It is stated that it is repeated "until a significant offset is found". But once an offset is found there could be others

**True. The objective is to remove all non-significant offsets. The assumption is that if the offset with the smallest amplitude is found to be significant, then all others are significant (because they all have similar standard errors), and thus the tests can be stopped.**

and, since only evenly spaced arrays of offsets are tested in each iteration, there may be epochs that have not yet been tested. So how does the algorithm guarantee the completeness of the scan for a step at every epoch?

All possible epochs are not tested. We only test for potential offsets at fixed epochs (every 20 days). The assumption is that a real offset at any given epoch will be caught by the forced artificial offset located less than 10 days directly before or after. As such, we do not find the exact date of the real offset but its approximate date +/- 10 days. This method cannot resolve real offsets situated within a few (10-20) days of each other. They will be lumped into a single artificial offset, but we assume that its effect on the estimated velocity will be a good proxy of the combined effect of the real offsets.

Secondly, if a large true step exists and the adjacent epoch is tested, it will likely be evaluated as a significant step. Is there a mechanism to replace the adjacent epoch with the correct one once it has been tested?

**No, this is not included.**

When the process is repeated are steps and non-steps identified in previous iterations excluded from being considered as steps? If so I expect that would improve efficiency and reduce ambiguity in the algorithm.

At each iteration, the fixed date of the non-significant offset found in the previous iteration is removed. All other remaining dates are kept, offsets at these dates are resolved, and the smallest is tested for significance.

Finally, could this method be applied to real data? It seems that the calibration method determining b and delta\_t in Appendix B relies on the quantity of false and true identifications, so might not be available for real data(?).

**Indeed, the calibration is not possible on real data.**

Altogether, it is important to keep in mind that this method was only developed as a simple and quick way to test the impact of offsets and their resolution of the velocity estimations. Our tests on synthetic data allow us to show that the method works, statistically as well as others, but we did not try to fine-tune or improve it.

Page 9 line 21. "real" need not be in quotes.

Indeed, we made the necessary changes. Thank you.

Page 9 line 27. "is no the case" should be "is not the case"

**Indeed, we made the necessary changes. Thank you again. (P10L17)**

Page 10 line 14-22. These classifications may be useful but possibly a bit dismissive of the utility of some of the categories when signals are large enough to stand out from the noise. For example the Oregon coast rises >4 mm/yr owing to elastic strain accumulation on the subduction zone. Even short time series may be useful there.

**True. Our implicit purpose here was to classify the GPS velocities for applications that require sub mm yr-1 precision. We rephrased this section better explain this and not imply a generic application of this classification.**

Page 1 Line 15: ... less than 4.5 years are not suitable for studies that require sub mm yr-1 precisions; (3) Series of intermediate ...

Page 11 Line 4: ... that may be applicable to actual GPS data used for high-precision (sub mm yr-1) studies, considering the fact that series duration is the key parameter ...

Page 14 Line 20: ... cannot be used for application that require a precision better than 1.0 mm yr-1, except ...

Page 10 line 31. "These results may indicate a lower limit in velocity accuracy ca. 0.1 mm yr-1". But it said in the previous sentence that some were 0.05 mm yr-1...

**True. The ambiguity stems from the fact that 0.05 mm $yr^{-1}$ is associated with noise-alone series (no offset), whereas the following sentence gives a more general value of 0.1 mm $yr^{-1}$ derived from the generic cases (noise + offsets). This is clarified by removing the number:**

Page 11 Line 20: ... noise effect as a function of time stagnates ca. 15 to 21 years (cf. Fig. 4 and section 3.1). Our results may indicate an overall lower limit on the velocity bias ca. 0.1 mm yr-1 due ...

Page 11 line 13. "v" should be "v\_95"? Also "A ratio of 1 corresponds to a standard error equal to its velocity". In a Gaussian distribution +/- one standard deviation contains 68% of the samples, whereas the definition of v\_95 in this paper is the limit that contains 95%. So would not a ratio of 2 indicate that the standard error and velocity accuracy are similar?

This section (and the associated figure caption) was not clear. We do not compute the ratio of v95 over a standard error (because v95 is the 95% quantile of the whole dataset), but rather the ratio of each estimated velocity over its standard error (for each individual time series). Thus, ratios of 1 and 2 should correspond to 68% and 95% of the populations. We clarify this in the text:

Page 12 Line 3: ... We can test the robustness of these standard errors in comparison with their associated velocity biases by computing the ratio of the velocity bias to its standard error for each individual time series. A ratio of 1 corresponds to a standard error equal to its velocity bias; a ratio smaller (greater) than 1 corresponds to a standard error greater

(smaller) than its velocity bias. Owing to our stochastic approach, and assuming Gaussian distributions of the velocities and standard errors, appropriate standard error calculations should result in ca. 68% of the ratio population smaller than 1 (i.e., 68% of the velocity biases are included in their standard errors) and ca. 95% of the population smaller than 2 (i.e., 95% of the velocity biases are included in twice their standard errors). In our dataset, only 54% of the ratio are smaller than 1 and 75% are smaller than 2 (Fig. 8). ...

Page 12 line 14. I did not see Masson et al., 2018 in the reference list.

**Yes, thank you.**

Page 13 line 28. "A significant outcome of our analysis is the fact that very long series durations (over 15 - 20 years) do not ensure a better accuracy compare to series with 8 - 10 years of measurements". However, Figure 4 says they are still getting more precise even at 20 years (though apparently at a decreasing rate of improvement) so I'm not sure if this statement is strictly true. It may be true that if a specific requirement for uncertainty is 0.1 mm/yr then there is no need to collect longer time series, but that requirement standard depends on the application and we may not yet know all future standards that are needed from the data.

**Yes I understand. It's not clear.**

Page 14 Line 25: A significant outcome of our analysis is that, beyond 8 years of data, it is the presence of offsets and the noise level that have the greatest impact on the velocity bias, and not the lengthening of the series (within the limit of 21 years tested here). This suggests that the lengthening of the series is not a sufficient condition to significantly reduce the bias in estimated velocities (below the 0.1 mm yr-1 level). This effect derives directly from our noise model definition, in which the noise amplitude follows a linear power-law dependency on the frequency (Eq. 2). As a result, the noise amplitude constantly increases with long periods, explaining the very small effect of the time series duration past ca. 10 years (cf. Fig. 4). Alternative noise models, such as Gauss-Markov, that predicts a flattening of the power spectrum at long periods would likely change our results and reinstate a strong duration dependency for very long series. This shows the importance of better characterization of the GPS noise nature at very long periods and of current efforts to model and correct for long-period signals such as pluri-annual environmental loads.

Page 14 line 5. Acknowledgements sections often now contain proper attribution to those who collected (in this case the RENAG network), archived, processed the data, and from where the processed time series were downloaded, i.e ftp server, web site, etc., and on what date. In this case the authors may have had prior access to the data (?), i.e. processed it themselves, but it would improve repeatability of this work if others could be guided to where they could access the data.

**We added a "Data Availability section" to address this point and the next:**

Page 15 Line 3

7 Data availability

The synthetic datasets and statistical analyses were performed using R (R Core Team, 2016). The synthetic time series dataset is available upon request to the authors. Figure 9 was done with GMT5 (Wessel et al., 2011). RENAG RINEX GPS data are available from the RESIF-RENAG (RESIF., 2017). RENAG GPS data were processed using the CCRS-PPP software (cf. Nguyen et al., 2016; Masson et al., 2018, for processing details).

**Acknowledgments**

We are grateful to Pr. Gilles Ducharme (IMAG, U. Montpellier) for his critical help with the regression tree analysis. We thank Simon Williams and William Hammond for their reviews that improved the quality of this manuscript.

Separate questions: Are the synthetic time series developed here openly available?

**Yes, cf. addition to the new "Data availability" section above**

Page 25, line 5. Why not show b=10, discussed in the text, on the plot?

Yes, I understand. It's not clear. By decreasing b, the number of false detections explodes. By doing a test on a part of the dataset we saw that it was not useful to do it on the whole. So as this test was not done on the entire dataset we preferred not to include it in the figure. However this allowed a potential improvement of the method which I hope will be discussed in a specific article for this method of detection of offsets.

Page 26 Line 5: For reference, partial tests with b = 10 showed a dramatic increase of false detections, so we decided not to apply it to the entire dataset.

**Precision of continuous GPS velocities from statistical analysis of synthetic time series**

Christine Masson1, Stephane Mazzotti1, Philippe Vernant1

1 Géosciences Montpellier, CNRS, University of Montpellier, Université des Antilles, Montpellier, 34000, France

Correspondence to: Christine Masson (masson@gm.univ-montp2.fr)

Abstract. We use statistical analyses of synthetic position time series to estimate the potential precision of GPS velocities. The synthetic series represent the standard range of noise, seasonal, and position offset characteristics, leaving aside extreme values. This analysis is combined with a new simple method for automatic offset detection that allows an automatic treatment of the massive dataset. Colored noise and the presence of offsets are the primary contributor to velocity variability.

- However, regression tree analyses show that the main factors controlling the velocity precision are first the duration of the series, second the presence of offsets, and third the noise level (dispersion and spectral index). Our analysis allows us to propose guidelines, which can be applied to actual GPS data, that constrain velocity precisions, characterized as a 95% confidence limits of the velocity biases, based on simple parameters: (1) Series durations over 8.0 years result in low
- 15 velocity biases in the horizontal (0.2 mm yr-1) and vertical (0.5 mm yr-1) components; (2) Series durations of less than 4.5 years are not suitable for studies that require sub mm yr-1 precisions; (3) Series of intermediate durations (4.5 8.0 years) are associated with an intermediate horizontal bias (0.6 mm yr-1) and a high vertical one (1.3 mm yr-1), unless they comprise no offset. Our results suggest that very long series durations (over 15 20 years) do not ensure a significantly lower bias compare to series of 8 10 years, due to the noise amplitude following a power-law dependency on the frequency. Thus,
- 20 better characterizations of long-period GPS noise and pluri-annual environmental loads are critical to further improve GPS velocity precisions.

**1 Introduction**

5

10

GPS (Global Positioning System) and more recently GNSS (Global Navigation Satellite System) have become classical datasets to study present-day tectonics, from active plate boundary regions (e.g., Serpelloni et al., 2013; McClusky et al.,

25 2000 to intraplate domains (e.g., Frankel et al., 2011; Tarayoun et al., 2018). GPS data processing, and thus the associated precision of GPS velocities have significantly improved in the last 20 years owing, for example, to the contribution of studies on noise characteristics (Williams et al., 2003), ionospheric effects (Petrie et al., 2010) or multipath and geometry effects (King et al., 2010). However, several state of the contracteristics of GPS velocities require that the velocities be defined with increasingly better precisions, potentially as low as 0.1 mm\_yr-1 or better. Typical examples of such

[revised manuscript text omitted]

|------------------------------------------------------------------------------------------------------------------------------------------------------------------------------------------------------------------------------------------------------------------------------------------------------------------------------------------------|
| Christine Masson 1/23/y 18:35                                                                                                                                                                                                                                                                                                                  |
| Christine Masson 1/23/y 18:35                                                                                                                                                                                                                                                                                                                  |
| Christine Masson 1/23/y 18:36                                                                                                                                                                                                                                                                                                                  |
| Christine Masson 1/23/y 18:37                                                                                                                                                                                                                                                                                                                  |
| Christine Masson 1/23/y 18:38                                                                                                                                                                                                                                                                                                                  |
| Christine Masson 1/23/y 18:39                                                                                                                                                                                                                                                                                                                  |
| Christine Masson 1/23/y 18:39                                                                                                                                                                                                                                                                                                                  |
| Christine Masson 1/23/y 18:40                                                                                                                                                                                                                                                                                                                  |
the 95 percentile (i.e., the 95% confidence limit in
the velocity accuracy), and $p_{al}$ , the percentile
associate with a velocity accuracy of 0.1 mm yr -1 .                                                                                             |
| Christine Masson 1/23/y 18:41                                                                                                                                                                                                                                                                                                                  |
| Christine Masson 1/23/y 18:41                                                                                                                                                                                                                                                                                                                  |
| Christine Masson 1/23/y 18:41                                                                                                                                                                                                                                                                                                                  |
| Christine Masson 1/23/v 18:42                                                                                                                                                                                                                                                                                                                  |

series confirm this hypothesis by yielding velocity biases ca.  $0.01 \text{ mm yr}^{-1}$  for the shortest series (< 4 yr) and smaller than 0.01 mm yr-1 in all other cases, including any of the three combinations of annual and semi-annual seasonal terms. Thus, in the following we focus on the effect of colored noise alone and colored noise with offsets, which are the main contributors to the velocity uncertainties.

**5 3.1 Effect of colored noise**

In order to estimate the impact of colored noise alone, we construct synthetic series using a subset of Eq.1:

$$x(t) = vt + D.rand(k,t)$$
(4)

- 10 We first analyze the effect of the three parameters the duration of the series (*T*), the spectral index (*k*) and the noise dispersion (*D*) - independently of the others. Figure 4 shows the velocity biases as a function of these three parameters. The worst values of velocity bias due to noise alone can reach  $v_{95} = 0.7$  mm yr-1 for the shortest series (T < 5 yr). For series longer than 15 years, all  $v_{95}$  are smaller than 0.1 mm yr-1. A near-exponential decrease of  $v_{95}$  is observed as a function of the duration of the series with a sharp slowdown from 15 years of data. The dependence of the velocity biases on noise
- 15 parameters (k and D) shows an expected bias increase with smaller spectral indices (closer to -1) and higher noise amplitudes, with a near-exponential increase with D. Overall, the probability of velocity biases equal to or smaller than 0.1 mm yr-1 is  $p_{01} = 86\%$ . The 14% of series with biases larger than 0.1 mm yr-1 are associated with the shortest and noisiest series.
- A joint analysis of the parameters using a regression tree indicates their relative importance, with the most important being
  the series duration *T* (56%) followed by the spectral index *k* (35%) and the noise dispersion *D* (9%). Figure 5 shows the tree classification (Fig. 5a) and the whisker plots of the associated leaves (Fig. 5b). The branches and the associated leaves are ordered in order of importance and leaf size from left to right. The comparison signs (> <) or (<>) are relative to each tree separation, with the sign on the left corresponding to the left branch and the sign on the right branch. Hereafter, we limit the tree classification to three node levels in order to only highlight the primary controlling
  elements.

The tree classification shows that  $y_{95} = 0.1 \text{ mm yr}^{-1}$  is achieved for over 2/3 of the series (leaves 1 and 2) corresponding to all the long series (T > 11.0 yr, leaves 1 and 2) and those with average durations and large spectral indices (6.1 < T < 11.0 yr, k > -0.6, leaf 1). The overall velocity bias increases for the other leaves.  $y_{95} = 0.2 \text{ mm yr}^{-1}$  is still reached for combinations of average durations and small spectral indices (6.1 < T < 11.0 yr, k < -0.6, leaf 3) or short durations, large spectral indices and

30 low noise amplitude (T < 6.1 yr, k > -0.7, D < 2.6 mm, leaf 4). The remaining cases (short duration, small spectral index, high noise) represent less than 10% of the samples and result in Jarge biases with  $v_{95} = 0.4 \text{ mm yr}^{-1}$  (leaf 5) and  $v_{95} = 0.7 \text{ mm}$ yr-1 (leaf 6). Christine Masson 1/23/y 18:42 **Supprimé:** accuracies of Christine Masson 1/23/y 18:42 **Supprimé:** better

| 1                | Christine Masson 1/23/y 18:43                                                                                                                                                                                                                                                                                    |
|------------------|------------------------------------------------------------------------------------------------------------------------------------------------------------------------------------------------------------------------------------------------------------------------------------------------------------------|
| -                | Christine Masson 1/23/y 18:43                                                                                                                                                                                                                                                                                    |
| 1                | Christine Masson 1/23/y 18:44                                                                                                                                                                                                                                                                                    |
|                  | Christine Masson 1/23/y 18:44                                                                                                                                                                                                                                                                                    |
| $\left( \right)$ | Christine Masson 1/23/y 18:44                                                                                                                                                                                                                                                                                    |
| $\left( \right)$ | Christine Masson 1/23/y 18:44                                                                                                                                                                                                                                                                                    |
|                  | Christine Masson 1/23/y 18:45                                                                                                                                                                                                                                                                                    |
|                  | Christine Masson 1/23/y 18:45                                                                                                                                                                                                                                                                                    |
|                  | Christine Masson 1/23/y 19:48                                                                                                                                                                                                                                                                                    |
|                  | Christine Masson 1/23/y 18:45                                                                                                                                                                                                                                                                                    |
|                  | Christine Masson 1/23/y 18:46                                                                                                                                                                                                                                                                                    |
|                  | Christine Masson 1/23/y 18:47                                                                                                                                                                                                                                                                                    |
|                  | Christine Masson 1/23/y 18:47                                                                                                                                                                                                                                                                                    |
|                  | Christine Masson 1/23/y 18:47                                                                                                                                                                                                                                                                                    |

Additionally, a significant piece of information emerging from the regression tree analysis is the relatively low coefficient of determination  $R^2 \sim 0.5$ , which indicates that the combinations of the three model parameters (T, k, D) only explain about 50% of the dispersion in velocity biases. This points out the strong effect of the stochastic noise generation, which alone accounts for about half of the velocity variability. In other words, for a given set of parameters, the generated time series will

5 show variable characteristics (noise structures) that randomly impact the velocity estimations. We illustrate this point by estimating the dispersion of velocity biases for a sample of 300 series with constant parameters T = 10 yr, k = -0.7, D = 3.0mm (belonging to leaf 3 of the tree). The estimated velocities show a RMS dispersion of 0.2 mm yr-1, of the same order as dispersion observed in leaf 3 (Fig. 5b). This effect is more important, if the series is short.

Christine Masson 1/23/y As noted in the introduction to Section 3, seasonal signals have very little effect on the velocity estimations. This is also true Supprimé: the velocity Christine Masson 1/23/y Supprimé: all the Supprimé: as

10 for seasonal signals added to series with random noise, which yield similar results to those presented above for noise alone (e.g.,  $p_{01} = 86\%$ ) with the seasonal parameters (A1/2 combinations) ranking with negligible importance in the tree classification (less than 1%).

**3.2 Effect of offsets**

In order to test and estimate the effect of position offsets on velocity estimations, we analyze synthetic time series that 15 include offsets added to seasonal signal and random noise (Eq. 1). This choice is justified by the very low effect of offsets

- alone (cf. introduction of Section 3) and the fact that this combination is representative of real data, thus providing useful estimations of the expected precision of actual velocities. In the case of real data, dealing with offsets requires either fixing their dates (from equipment logs or earthquake catalogs) or detecting their potential occurrences. In section 4, we will come back to how to consider the latter. In this section, we quantify the two end-member cases in which we either do not know
- 20 and therefore do not solve any offset, or we know and solve all of them.

**Effect of unresolved offsets 3.2.1**

In this first simple case, we test time series with a single offset that is not solved, and quantify the importance of the offset parameters (amplitude  $C_l$  and position in the series  $t_{i}$  in addition to the parameters T, k, and D considered previously. A regression tree analysis indicates that the velocity variability is primarily controlled by the time series duration T (importance

25 49%), as in the case of noise alone, followed closely by the amplitude of the offset (40%). The position of the offset (5%), the noise amplitude (3%) and spectral index (3%) rank in 3rd, 4th, and 5th positions far beyond the two main parameters. The coefficient of determination is larger than for the noise alone ( $R^2 = 0.8$ ), indicating that the inclusion of a single offset contributes significantly to the overall velocity variability.

This is illustrated in Figure 6, which shows a distribution of velocity biases much larger than for the noise alone (cf. Fig. 4),

with  $v_{95}$  systematically above ca. 0.3 mm yr-1. The presence of a single unresolved offset increases  $v_{95}$  to 0.5 mm yr-1 for long 30 series (T > 13 yr) and up to 2.5 mm yr-1 for short series. Only about 1/5 of the series are associated with velocity biases below 0.1 mm yr-1 ( $p_{01} = 18\%$ , compare to  $p_{01} = 84\%$  for noise-alone series). As expected, the position of the offset in the

Christine Masson 1/23/y 18:49

Christine Mas Supprimé: T1

series has a significant impact, with an offset placed at one end of the series causing a velocity bias much lower than an offset placed in the central part.

In a second series of tests, we include, but do not solve, several offsets (between 0 and 7 offsets depending on the series length, cf. Section 2). In this case, we cannot quantify the impact of the amplitudes and positions of the offsets as single

- 5 parameters; instead we use the ratio of the number of offsets to the series duration T, which illustrates the proportion of offsets in the series. A regression tree analysis indicates the following parameter importance; T (53%), ratio of number of offsets to T (44%), D (2%) and k (1%), similar to the case of a single offset discussed above. About 2/3 of the series are associated with velocity piases below 0.1 mm yr-1 ( $p_{01} = 67\%$ ). The largest velocity piases occur on the shortest series. Uncorrected offsets are therefore a dominant element in the determination of the velocity. These conclusions on the role of
- 10 the position and magnitude of the offsets in the time series are consistent with the analytical analysis in Williams (2003b).

**3.2.2 Effect of resolved offsets**

As in the previous section, we first analyze the simple case of a series with one offset, but for which we fix the date and solve for the amplitude during the inversion. Thus, the velocity biases are affected by the possible imperfection of the estimated amplitude of the offsets, primarily due to the series colored noise. The regression tree analysis indicates that, when

- 15 the offset amplitude is solved, the offset parameters become of very low importance (amplitude and position at 2% each), while the series duration and noise parameters recover the same importance and order as in the case of noise alone: T 52%, k31% and D 13% (cf. section 3.1). The regression tree and associated velocity bias statistics are similar to that of the noise alone analysis (cf. Fig. A in appendix).  $v_{95}$  of all tree leaves are approximately 3 times lower than in the case of an unresolved offset but slightly larger than in the case of noise alone, in particular for short series.
- 20 Considering series with a variable number of offsets, for which we fix the date and solve for the amplitude, the importance of the parameters becomes intermediate between the noise-alone and single-offset cases: T 42%, ratio of number of offsets to T 21%, k 20% and D 17%. Resolving the offset amplitudes reduces their importance (21 % vs. 44%) but their presence remains a significant source of velocity variability, contrary to the case of a single solved offset by series. This is readily explained by the fact that the offset amplitudes are not perfectly resolved due to complex interaction between the offset
- 25 positions, their amplitudes, and the noise structure that result in potentially very short linear segments in the series. This is illustrated by the probabilities of biases lower than 0.1 mm yr-1 ( $p_{01} = 71\%$ ), slightly lower than in the case of noise only series ( $p_{01} = 86\%$ ).

This latest result represents the lower bounds of velocity biases for series with several offsets, assuming that all offset dates are know. In reality, we do not know the exact nature and dates of all potential offsets (e.g., Gazeaux et al., 2013), so it is

30 necessary to detect them before solving for their amplitude. In the next section, we propose a new detection method and test its impact on velocity biases. Christine Masson 1/23/y 18:52 Supprimé: deviation

| Christine Masson 1/23/y 18:54                  |
|------------------------------------------------|
| Christine Masson 1/23/y 18:54                  |
| Christine Masson 1/23/y 18:54                  |
| Christine Masson 1/23/y 18:55                  |
| Christine Masson 1/23/y 18:55                  |
| Christine Masson 1/23/y 18:55                  |
| Christine Masson 1/23/y 18:56                  |
|                                                |
| Christine Masson 1/23/y 18:57                  |

Christine Masson 1/23/y 18:5 Supprimé: accuracies

**4 A new approach for offset detection and impact on velocity bias**

**4.1 Methodology of offset detection**

Real GPS time series are associated with an indeterminate number of offsets, which are classically included as instantaneous changes of position in the series inversion, (cf. Eq. 1). Offset dates  $t_{j}$  can be based on equipment logs, catalogs of earthquakes, or routines that detect them in the position series (Gazeaux et al., 2013). Here we propose a slightly different

approach that does not consist of seeking where there are offsets, but rather in seeking where there are none.

This simple principle is implemented by defining artificial offset dates that are regularly spaced in the series every  $\Delta d$  days. The series is then inverted to estimate all offset amplitudes  $(\underline{C}_1)$  and their associated standard errors  $(\underline{\sigma}_{c1})$  jointly with the other model parameters (velocity, seasonal signal, etc.). The offset with the smallest amplitude  $(\underline{C}_S)$  is then identified and a simple significance test is performed;

 $amp_{off} \ge b * \sigma_{off}$  (5)

5

If the amplitude  $(\underline{C_{S}})$  is larger than its scaled standard error  $(\underline{b}, \sigma_{cs})_{\psi}$  the offset is considered significant. Because the test is

- 15 performed on the smallest offset and the offset standard errors are similar in the majority of cases, we then consider that all offsets are significant and we keep them in the model. In the opposite case, the smallest offset is rejected and the inversion is redone with the remaining offsets in order to test the new smallest offset, until a significant offset is found or none remains. This very simple approach can be implemented in most time series analysis and only requires an empirical calibration of the two parameters  $\Delta d$  and b. After several tests, we set the former to  $\Delta d = 20 \text{ days}$ , which corresponds to the lower limit before
- 20 the method breaks down (i.e., too many undifferentiated offsets). The latter is set to b = 20, which allows a good compromise between the detection of real offsets defined in the synthetic series and the detection of false positives (cf. Section 4.2). Details on the parameter calibration and the detection levels are available in appendix B.

**4.2 Detection ability**

By applying our method to series with only one offset, it is possible to determine the conditions of offset detections. Overall, 25 67% of the offsets are detected. The detection capacity depends primarily on the duration of the time series *T*, combined with the series noise amplitude *D* and the offset amplitude *C*. For the shortest time series (T < 6 yr), we detect 21% of offsets. They correspond to the series with the largest offsets (C > 3.0 mm) and the smallest noise amplitudes (D < 2.1 mm). There is no offset detection in the series with large noise amplitude (D > 2.1 mm).

For the time series of 6 to 18 years, we can detect offsets of small amplitudes (C = 1.0 - 3.0 mm) in series with low noise 30 levels (D < 2.1 mm) and large offsets (C > 4.0 mm) in all series. For the longest time series of more than 18 years, one widens still the range of detection. Offsets larger than 3.0 mm are systematically detected and those between 2.0 and 3.0 mm are detected at 49%. The very small offsets (C < 2.0 mm) are detected only in the low noise series (D < 2.1 mm).

**Christine Masson 1/23/y 18:5 Supprimé:**

**Christine Masson 1/23/v 18:5**

| Supprin
$C_i \cdot \delta(t, T)$
time of the | né: cf. Eq. 1 where offsets are modeled as
() where $C_i$ and $T_i$ are the amplitude and
he i th offset (with $\delta$ the Kronecker function) |
|----------------------------------------------------|-------------------------------------------------------------------------------------------------------------------------------------------------------------------------|
| Christir                                           | e Masson 1/23/y 18:58                                                                                                                                                   |
| Supprir                                            | né: The dates T i                                                                                                                                            |
| Christir                                           | e Masson 1/23/y 18:59                                                                                                                                                   |
| Supprir                                            | né: detects offsets                                                                                                                                                     |
| Christir                                           | e Masson 1/23/y 18:59                                                                                                                                                   |
| Supprin                                            | né: s                                                                                                                                                                   |
| Christir                                           | ne Masson 1/23/y 18:59                                                                                                                                                  |
| Supprin                                            | né: in                                                                                                                                                                  |
| Christir                                           | ne Masson 1/23/y 18:59                                                                                                                                                  |
| Supprin                                            | né: (amp off )                                                                                                                                               |
| Christir                                           | ne Masson 1/23/y 19:00                                                                                                                                                  |
| Supprin                                            | né: (σ off )                                                                                                                                                 |
| Christir                                           | ne Masson 1/23/y 19:00                                                                                                                                                  |
| Supprin
performe                                | né: A simple significance test is then d on the offset of smallest amplitude                                                                                     |
| Christir                                           | ne Masson 1/23/y 19:01                                                                                                                                                  |
| Supprin                                            | né: ,                                                                                                                                                                   |
| Christir                                           | ne Masson 1/23/y 19:01                                                                                                                                                  |
| Supprin
standard                                | né: (amplitude smaller than the scaled error)                                                                                                                    |
| Christir                                           | e Masson 1/23/y 19:01                                                                                                                                                   |
| Supprin
smallest                                | né: the model is rerun to test the new offset, until a significant offset is found.                                                                              |
| Christir                                           | e Masson 1/23/y 19:02                                                                                                                                                   |
| Supprin                                            | né: method                                                                                                                                                              |
| Christir                                           | ne Masson 1/23/y 19:02                                                                                                                                                  |
| Supprin                                            | né: simply                                                                                                                                                              |
| Christir                                           | ne Masson 1/23/y 19:02                                                                                                                                                  |
| Supprin                                            | né: d                                                                                                                                                                   |

5 Gazeaux et al., (2013). Although not perfect, our method allows us to obtain robust and quantitative results, and is suitable for processing of very large datasets such as our synthetic series or regional and global massive processing efforts that become increasingly common (e.g., Kreemer et al., 2014) and that could not be analyzed "by hand".

**4.3 Impact on the determination of the velocities**

The application of the offset detection method on a full dataset with multiple offsets, variable noise and seasonal signals provides a sample that can be considered as close as possible to actual GPS data. We use this analysis to provide constraints

- on the potential velocity precision in real data. Overall, nearly 2/3 of series are associated with velocity bias smaller than 0.1 mm yr-1 ( $p_{01} = 61\%$ ). This is lower than in the cases of noise alone ( $p_{01} = 86\%$ ) or fully resolved offsets ( $p_{01} = 71\%$ ), but significantly better than in the case of unresolved offsets ( $p_{01} = 33\%$ ). The difference between the results of the offset detection method and those of the fully resolved offsets (ca. 10%) is mainly associated with undetected offsets in the former.
- 15 For the regression tree analysis, the integration of a parameter associated with offsets is complex. Although these parameters (numbers total of offsets, of true and false detections, positions in the series, amplitudes) are known in our synthetic data,
  this is not the case in real datasets. Tests on several offset parameters indicate that the total number of offsets in the series (Noff) is both the simplest and the one with the highest prediction capacity. This new regression tree (Fig. 7) confirms the major role of the series duration (T 55%) and noise dispersion (D 16%) in explaining the variability of the velocities, but the
- 20 total number of offsets now take the second position ( $N_{off}$  25%), above the noise dispersion. It is particularly worth noting that the number of offsets is in fact a binary predicator (splitting value  $N_{off} = 0.5$ ) corresponding to either the absence ( $N_{off} = 0$ ) or the presence ( $N_{off} \ge 1$ ) of offsets in the series. To first order, the regression tree results can be divided in three categories:

The Jowest velocity biases ( $v_{95} \sim 0.2 - 0.3 \text{ mm yr}^{-1}$ ) are associated with either long (T > 8.0 yr) and low noise dispersion (D

- 25 < 2.3 mm) series, or with series of intermediate duration (4.5 < T < 8.0 yr) with no offset (leaves 1 and 3). These represent over 42% of the dataset.
  - Intermediate biases  $(v_{95} \sim 0.5 0.6 \text{ mm yr}^{-1})$  are associated with series characterized by long duration and high dispersion series (leaf 2), intermediate duration and low dispersion (leaf 4), or short duration (T < 4.5 yr) but no offset (leaf 6). Altogether, these represent another 43% of the dataset.
- The remainder ca. 15% correspond to high biases (v95 > 1.0 mm yr-1) and is mostly associated with short durations (leaves 7, 8, 9), or intermediate duration and high dispersion (leaf 5).

Tree nodes associated with the series dispersion D indicate that a systematic separation can be made at D = 2.2 - 2.3 mm (Fig. 7a). As shown in Figure 2, the separation between horizontal and vertical component dispersion occurs ca. D = 2.5 mm,

10

| 1 | Christine Masson 1/23/y 19:04 |
|---|-------------------------------|
|   | Christine Masson 1/23/y 19:04 |
|   | Christine Masson 1/23/y 19:04 |
|   | Christine Masson 1/23/y 19:05 |
|   | Christine Masson 1/23/y 19:05 |
|   | Christine Masson 1/23/y 19:05 |
|   | Christine Masson 1/23/y 19:06 |

|   | Christine Masson 1/23/y 19:07 |
|---|-------------------------------|
|   | Christine Masson 1/23/y 19:07 |
| 1 | Christine Masson 1/23/y 19:07 |

On these bases, a fairly simple set of rules can be derived from the regression tree analysis that may be applicable to actual GPS data used for high-precision (sub mm yr-1) studies, considering the fact that series duration is the key parameter:

Duration of 8.0 years or more ensures a low velocity bias in both horizontal ( $v_{95} = 0.2 \text{ mm yr}^{-1}$ ) and vertical ( $v_{95} = 0.5 \text{ mm yr}^{-1}$ ) components.

Short series with less than 4.5-years duration cannot be used for high-precision studies ( $v_{95} > 1.0 \text{ mm yr}^{-1}$ ), except in the rare cases when one can be certain that they contain no significant offset.

For intermediate durations (4.5 < T < 8.0 yr), only series with no offset can provide a low velocity bias ( $v_{95} = 0.3 \text{ mm yr}^{-1}$ ). All others are associated with an intermediate horizontal biases ( $v_{95} = 0.6 \text{ mm yr}^{-1}$ ) and a high vertical one ( $v_{95} = 1.3 \text{ mm yr}^{-1}$ ).

The strong dependency on the absence or presence of one or more offsets in intermediate and short series corresponds to the effect described in Section 3.2 and confirms that the resolution of the offset amplitude is limited by the complex interactions

15 between offsets and noise structures. This effect is very strongly reduced (or possibly suppressed) when offsets affect long (T > 8.0 yr) series. For those, the velocity variability is independent of offset presence (Fig. 7a) because such series maintain relatively long "offset free" segments that ensure a good resolution of the velocity.

Finally, it is significant that no tree node exists that distinguishes very long series. In other words, the effect of the series duration is limited to ca. 4.5 yr and 8.0 yr. This is consistent with the observation made in the noise-alone analysis that the

- 20 decay of the noise effect as a function of time stagnates ca. 15 to 21 years (cf. Fig. 4 and section 3.1). Our results may indicate an overall lower limit on the velocity bias ca. 0.1 mm yr-1 due to the colored nature of the time series noise. In other words, longer series may not be able to significantly reduce the velocity bias without additional efforts to whiten the noise through better data processing or taking into account pluri-annual signals. However, this hypothesis is only valid under the simple noise model (linear spectra, Eq. 2) used in our synthetic data. Alternative noise models exist that suggest a flattening
- 25 of the spectra at long periods (e.g., Gauss-Markov model, Langbein et al., 2004), which would strongly limit the pluri-annual effect and allow a much stronger impact of long series duration. The actual nature of GPS noise at periods longer than 5 10 years is poorly defined (Santamaria-Gomez et al., 2011; Hackl et al., 2011) and is thus a major unknown in analyses of velocity precision.

**4.4 Validation of velocity standard errors**

30 For each series, the velocity standard error is calculated using Williams (2003) generic expression for colored noise with non-integer spectral index. In order to estimate the spectral index and amplitude of the colored noise, we use a simplified least-square inversion in which we fit a linear model to the series power spectrum limited to periods between 1/12 and T/2 years (with T the length of the time series). In contrast with a more complex non-linear method, such as maximum

11

| Christine Masson 1/23/y 19:10 |
|-------------------------------|
| Christine Masson 1/23/y 19:10 |
| Christine Masson 1/23/y 19:11 |
| Christine Masson 1/23/y 19:11 |
| Christine Masson 1/23/y 19:11 |
| Christine Masson 1/23/y 19:11 |

|                   | Christine Masson 1/23/y 19:11                      |
|-------------------|----------------------------------------------------|
|                   | Christine Masson 1/23/y 19:11                      |
|                   | Christine Masson 1/23/y 19:12                      |
| $\left( \right)$  | Christine Masson 1/23/y 19:12                      |
| $\left( \right)$  | Christine Masson 1/23/y 19:12                      |
| $\left( \right)$  | Christine Masson 1/23/y 19:12                      |
|                   | Christine Masson 1/23/y 19:13                      |
|                   | Christine Masson 1/23/y 19:13                      |

likelihood, this simple approach does not solve for the noise crossover frequency and thus only provides a first-order estimate of the noise parameters and velocity standard errors.

We can test the robustness of these standard errors in comparison with their associated velocity biases by computing the ratio of the velocity bias to its standard error for each individual time series. A ratio of 1 corresponds to a standard error equal to

- 5 its velocity bias; a ratio smaller (greater) than 1 corresponds to a standard error greater (smaller) than its velocity bias. Owing to our stochastic approach, and assuming Gaussian distributions of the velocities and standard errors, appropriate / standard error calculations should result in, ca. 68% of the ratio population smaller than 1 (i.e., 68% of the velocity biases are included in their standard errors) and ca. 95% of the population smaller than 2 (i.e., 95% of the velocity biases are included in twice their standard errors). In our dataset, only 54% of the ratio are smaller than 1 and 75% are smaller than 2 (Fig. 8).
- 10 These percentages are low and suggest that, on average, our velocity standard errors are too small by a factor ca. 1.6. This result is primarily controlled by the series spectral index, while the series duration and dispersion have little effect (Fig. 8). Series with indices ca. (-0.6 > k > -0.9) are associated with ratio percentages close to the 68 and 95% marks. In contrast, series with high indices (k > -0.6) present ratios that are too low especially for very high indices (k > -0.4). These results suggest that the simplified (linear spectra) approach yields reasonable results for series with near-flicker (k < -0.6) noise
- 15 characteristics, but significantly underestimates the standard errors for series with near-white  $(k \ge -0.4)$  noise.

**5 Application to the RENAG data**

The statistical analyses of synthetic data presented in the previous sections provide guidelines to estimate the precision of velocities from actual GPS data. Using the regression tree classification of the full synthetic dataset with automatic offset detection, (Section 4.3), actual time series can be classified according to the primary controlling parameters (duration,

20 presence of offsets, noise amplitude and spectral index) and associated with a velocity bias distribution (Fig. 7). In the following application to the French RENAG network (RESIF, 2017), we use the 95% confidence limit ( $v_{95}$ ) estimator to provide a measure of the velocity precision of these real data. This estimator can be viewed as the classical velocity "uncertainty at 95% confidence" (twice the standard error),

**5.1 Offsets due to equipement changes**

25 The RENAG network comprises 74 stations whose equipment modifications are fully documented (cf., http://webrenag.unice.fr), thus providing a good test case for our offset detection method. On the 222 time series with durations between 2.0 and 18.4 years, the comparison of detected offsets with the station logs show that a change of receiver is very rarely associated with an offset (only 6% of the 137 cases), whereas a change of antenna causes an offset almost systematically (75% of the 8 cases) with average amplitudes of 2.0 – 3.0 mm in the horizontal and ca. 13.0 mm in the

systematically (1576 of the 6 cases) with average ampiritudes of  $2.0^{\circ}$  5.0 min in the intribution and  $\frac{1}{2.0^{\circ}}$  is systematically (1576 of the 6 cases) with average ampiritudes of  $2.0^{\circ}$  5.0 min in the intribution and  $\frac{1}{2.0^{\circ}}$  is specially the antenna 30 vertical components, However, these percentages are not robust due to the small sample sizes (especially the antenna

changes). A more robust analysis would require a larger dataset, as well as the distinction between equipment changes within

| Ι | Christine Masson 1/23/y 19:15                                                                                                                                                                                                                                                                                                                                                                                                                                                                                                                                                                                                                                                                                                                                                                                                                                                                                                                                                                                                                                                                                                                                                                                                                                                                                                                                                                                                                                                                                                                                                                                                                                                                                                                                                                                                                                                                                                                                                                                                                                                                                                      |
|---|------------------------------------------------------------------------------------------------------------------------------------------------------------------------------------------------------------------------------------------------------------------------------------------------------------------------------------------------------------------------------------------------------------------------------------------------------------------------------------------------------------------------------------------------------------------------------------------------------------------------------------------------------------------------------------------------------------------------------------------------------------------------------------------------------------------------------------------------------------------------------------------------------------------------------------------------------------------------------------------------------------------------------------------------------------------------------------------------------------------------------------------------------------------------------------------------------------------------------------------------------------------------------------------------------------------------------------------------------------------------------------------------------------------------------------------------------------------------------------------------------------------------------------------------------------------------------------------------------------------------------------------------------------------------------------------------------------------------------------------------------------------------------------------------------------------------------------------------------------------------------------------------------------------------------------------------------------------------------------------------------------------------------------------------------------------------------------------------------------------------------------|
|   | time series to estimate their robustness                                                                                                                                                                                                                                                                                                                                                                                                                                                                                                                                                                                                                                                                                                                                                                                                                                                                                                                                                                                                                                                                                                                                                                                                                                                                                                                                                                                                                                                                                                                                                                                                                                                                                                                                                                                                                                                                                                                                                                                                                                                                                           |
| J | Christine Masson 1/23/y 19:16                                                                                                                                                                                                                                                                                                                                                                                                                                                                                                                                                                                                                                                                                                                                                                                                                                                                                                                                                                                                                                                                                                                                                                                                                                                                                                                                                                                                                                                                                                                                                                                                                                                                                                                                                                                                                                                                                                                                                                                                                                                                                                      |
|   | 1 corresponds to a standard error greater (smaller)                                                                                                                                                                                                                                                                                                                                                                                                                                                                                                                                                                                                                                                                                                                                                                                                                                                                                                                                                                                                                                                                                                                                                                                                                                                                                                                                                                                                                                                                                                                                                                                                                                                                                                                                                                                                                                                                                                                                                                                                                                                                                |
|   | than its velocity.                                                                                                                                                                                                                                                                                                                                                                                                                                                                                                                                                                                                                                                                                                                                                                                                                                                                                                                                                                                                                                                                                                                                                                                                                                                                                                                                                                                                                                                                                                                                                                                                                                                                                                                                                                                                                                                                                                                                                                                                                                                                                                                 |
| 1 | Christine Masson 1/23/y 19:16                                                                                                                                                                                                                                                                                                                                                                                                                                                                                                                                                                                                                                                                                                                                                                                                                                                                                                                                                                                                                                                                                                                                                                                                                                                                                                                                                                                                                                                                                                                                                                                                                                                                                                                                                                                                                                                                                                                                                                                                                                                                                                      |
|   | Christine Masson 1/23/y 19:17                                                                                                                                                                                                                                                                                                                                                                                                                                                                                                                                                                                                                                                                                                                                                                                                                                                                                                                                                                                                                                                                                                                                                                                                                                                                                                                                                                                                                                                                                                                                                                                                                                                                                                                                                                                                                                                                                                                                                                                                                                                                                                      |
|   | Christine Masson 1/23/y 19:17                                                                                                                                                                                                                                                                                                                                                                                                                                                                                                                                                                                                                                                                                                                                                                                                                                                                                                                                                                                                                                                                                                                                                                                                                                                                                                                                                                                                                                                                                                                                                                                                                                                                                                                                                                                                                                                                                                                                                                                                                                                                                                      |
|   | Christine Masson 1/23/y 19:17                                                                                                                                                                                                                                                                                                                                                                                                                                                                                                                                                                                                                                                                                                                                                                                                                                                                                                                                                                                                                                                                                                                                                                                                                                                                                                                                                                                                                                                                                                                                                                                                                                                                                                                                                                                                                                                                                                                                                                                                                                                                                                      |
|   | Christine Masson 1/23/y 19:18                                                                                                                                                                                                                                                                                                                                                                                                                                                                                                                                                                                                                                                                                                                                                                                                                                                                                                                                                                                                                                                                                                                                                                                                                                                                                                                                                                                                                                                                                                                                                                                                                                                                                                                                                                                                                                                                                                                                                                                                                                                                                                      |
|   | Christine Masson 1/23/y 19:51                                                                                                                                                                                                                                                                                                                                                                                                                                                                                                                                                                                                                                                                                                                                                                                                                                                                                                                                                                                                                                                                                                                                                                                                                                                                                                                                                                                                                                                                                                                                                                                                                                                                                                                                                                                                                                                                                                                                                                                                                                                                                                      |
|   | Christine Masson 1/23/y 19:51                                                                                                                                                                                                                                                                                                                                                                                                                                                                                                                                                                                                                                                                                                                                                                                                                                                                                                                                                                                                                                                                                                                                                                                                                                                                                                                                                                                                                                                                                                                                                                                                                                                                                                                                                                                                                                                                                                                                                                                                                                                                                                      |
|   | Christine Masson 1/23/y 19:18                                                                                                                                                                                                                                                                                                                                                                                                                                                                                                                                                                                                                                                                                                                                                                                                                                                                                                                                                                                                                                                                                                                                                                                                                                                                                                                                                                                                                                                                                                                                                                                                                                                                                                                                                                                                                                                                                                                                                                                                                                                                                                      |
|   | Christine Masson 1/23/y 19:19                                                                                                                                                                                                                                                                                                                                                                                                                                                                                                                                                                                                                                                                                                                                                                                                                                                                                                                                                                                                                                                                                                                                                                                                                                                                                                                                                                                                                                                                                                                                                                                                                                                                                                                                                                                                                                                                                                                                                                                                                                                                                                      |
| 1 | Christine Masson 1/23/y 19:19                                                                                                                                                                                                                                                                                                                                                                                                                                                                                                                                                                                                                                                                                                                                                                                                                                                                                                                                                                                                                                                                                                                                                                                                                                                                                                                                                                                                                                                                                                                                                                                                                                                                                                                                                                                                                                                                                                                                                                                                                                                                                                      |
| ١ | Christine Masson 1/23/y 19:19                                                                                                                                                                                                                                                                                                                                                                                                                                                                                                                                                                                                                                                                                                                                                                                                                                                                                                                                                                                                                                                                                                                                                                                                                                                                                                                                                                                                                                                                                                                                                                                                                                                                                                                                                                                                                                                                                                                                                                                                                                                                                                      |
|   | Christine Masson 1/23/y 19:19                                                                                                                                                                                                                                                                                                                                                                                                                                                                                                                                                                                                                                                                                                                                                                                                                                                                                                                                                                                                                                                                                                                                                                                                                                                                                                                                                                                                                                                                                                                                                                                                                                                                                                                                                                                                                                                                                                                                                                                                                                                                                                      |
|   | Christine Masson 1/23/y 19:20                                                                                                                                                                                                                                                                                                                                                                                                                                                                                                                                                                                                                                                                                                                                                                                                                                                                                                                                                                                                                                                                                                                                                                                                                                                                                                                                                                                                                                                                                                                                                                                                                                                                                                                                                                                                                                                                                                                                                                                                                                                                                                      |
|   | Christine Masson 1/23/y 19:20                                                                                                                                                                                                                                                                                                                                                                                                                                                                                                                                                                                                                                                                                                                                                                                                                                                                                                                                                                                                                                                                                                                                                                                                                                                                                                                                                                                                                                                                                                                                                                                                                                                                                                                                                                                                                                                                                                                                                                                                                                                                                                      |
|   | Christine Masson 1/23/y 19:51                                                                                                                                                                                                                                                                                                                                                                                                                                                                                                                                                                                                                                                                                                                                                                                                                                                                                                                                                                                                                                                                                                                                                                                                                                                                                                                                                                                                                                                                                                                                                                                                                                                                                                                                                                                                                                                                                                                                                                                                                                                                                                      |
|   | Christine Masson 1/23/y 19:21                                                                                                                                                                                                                                                                                                                                                                                                                                                                                                                                                                                                                                                                                                                                                                                                                                                                                                                                                                                                                                                                                                                                                                                                                                                                                                                                                                                                                                                                                                                                                                                                                                                                                                                                                                                                                                                                                                                                                                                                                                                                                                      |
|   | Christine Masson 1/23/y 19:22                                                                                                                                                                                                                                                                                                                                                                                                                                                                                                                                                                                                                                                                                                                                                                                                                                                                                                                                                                                                                                                                                                                                                                                                                                                                                                                                                                                                                                                                                                                                                                                                                                                                                                                                                                                                                                                                                                                                                                                                                                                                                                      |
|   | velocity precision (e.g., $v_{95}$ ) simply using the [2]                                                                                                                                                                                                                                                                                                                                                                                                                                                                                                                                                                                                                                                                                                                                                                                                                                                                                                                                                                                                                                                                                                                                                                                                                                                                                                                                                                                                                                                                                                                                                                                                                                                                                                                                                                                                                                                                                                                                                                                                                                                                          |
|   | Christine Masson 1/23/y 19:22                                                                                                                                                                                                                                                                                                                                                                                                                                                                                                                                                                                                                                                                                                                                                                                                                                                                                                                                                                                                                                                                                                                                                                                                                                                                                                                                                                                                                                                                                                                                                                                                                                                                                                                                                                                                                                                                                                                                                                                                                                                                                                      |
|   | Christine Masson 1/23/y 19:22